# Ultraviolet light blocking optically clear adhesives for foldable displays via highly efficient visible-light curing

Yonghwan Kwon[1,6], Seokju Lee[1,6], Junkyu Kim[1], Jinwon Jun [2], Woojin Jeon[1], Youngjoo Park[1], Hyun-Joong Kim[3], Johannes Gierschner [4], Jaesang Lee [2]✉, Youngdo Kim[5]✉ & Min Sang Kwon [1]✉

In developing an organic light-emitting diode (OLED) panel for a foldable smartphone (specifically, a color filter on encapsulation) aimed at reducing power consumption, the use of a new optically clear adhesive (OCA) that blocks UV light was crucial. However, the incorporation of a UV-blocking agent within the OCA presented a challenge, as it restricted the traditional UV-curing methods commonly used in the manufacturing process. Although a visible-light curing technique for producing UV-blocking OCA was proposed, its slow curing speed posed a barrier to commercialization. Our study introduces a highly efficient photo-initiating system (PIS) for the rapid production of UV-blocking OCAs utilizing visible light. We have carefully selected the photo-catalyst (PC) to minimize electron and energy transfer to UV-blocking agents and have chosen co-initiators that allow for faster electron transfer and more rapid PC regeneration compared to previously established amine-based co-initiators. This advancement enabled a tenfold increase in the production speed of UV-blocking OCAs, while maintaining their essential protective, transparent, and flexible properties. When applied to OLED devices, this OCA demonstrated UV protection, suggesting its potential for broader application in the safeguarding of various smart devices.

Smartphones have revolutionized the way we live, work, and interact with the world, making them an indispensable tool for communication, entertainment, and access to information for both personal and professional use. The recent emergence of smartphones with various form factors has further provided users with improved features such as a better viewing experience, increased durability, and enhanced portability[1].

The intricate functionality of a smartphone is achieved through the seamless integration of distinct films, each designed to fulfill specific functions. In this respect, an adhesive is one of the most crucial components of a modern smartphone[2,3]. With the emergence of smartphones featuring various form factors, adhesives must fulfill multiple requirements beyond connecting the individual element films[4–6]. Specifically, they need to provide stress dissipation properties while maintaining a suitable balance between flexibility and elasticity[7]. In addition, they should exhibit high optical transparency and exceptional stability under conditions of moisture and heat[3]. Consequently, a well-designed adhesive can maintain its structural integrity even in

[1]Department of Materials Science and Engineering, Seoul National University, Seoul, Republic of Korea. [2]Department of Electrical and Computer Engineering, Seoul National University, Seoul, Republic of Korea. [3]Department of Agriculture, Forestry and Bioresources, Seoul National University, Seoul, Republic of Korea. [4]Madrid Institute for Advanced Studies, IMDEA Nanoscience, Calle Faraday 9, Campus Cantoblanco, 28049 Madrid, Spain. [5]Samsung Display Co., Ltd., Cheonan, Republic of Korea. [6]These authors contributed equally: Yonghwan Kwon, Seokju Lee. ✉e-mail: jsanglee@snu.ac.kr; colour.kim@samsung.com; minsang@snu.ac.kr

extreme operating conditions that involve multiple deformations without exhibiting permanent deformation or failure.

Reducing power consumption is a crucial issue in smartphones for improving battery life, device performance, and lifespan, while minimizing environmental impact during manufacturing and charging. Indeed, recently, a state-of-the-art foldable smartphone has been introduced, featuring innovative technology that reduces the display panel's power consumption by 25% (i.e., color filter (CF) on encapsulation, Fig. 1a)[8]. The new panel design implemented in the current foldable display eliminates the requirement for an additional polarizer layer, which is typically employed to mitigate external light reflection. This innovative approach necessitates the modification of existing layers to possess UV-blocking characteristics to replace the polarizer's function of safeguarding the panel from external UV radiation. To incorporate a UV-blocking ability into the target layer,

its fabrication process needs to be compatible with additives which efficiently absorb UV radiation, such as pigments or UV absorbers (UVAs). One approach to introduce these capabilities would involve fine-tuning the pigments in the CF layer, but it would require endeavours to ensure they absorb UV light and visible light simultaneously at desired wavelengths[9–11]. An alternative strategy would be imbedding UVAs into the CF layer. However, since the fabrication of these filters typically relies on UV-light curing[12,13], substantial efforts are needed to overcome potential hindrances of UV-light curing associated with their high UV-light absorbance. Optically clear adhesives (OCA) layer could be regarded as the other alternative, but the preparation of UV-blocking adhesive still proves to be a complex task in the presence of UVAs, which impedes the effectiveness of the conventional photocuring method that relies on a UV-photoinitiator.

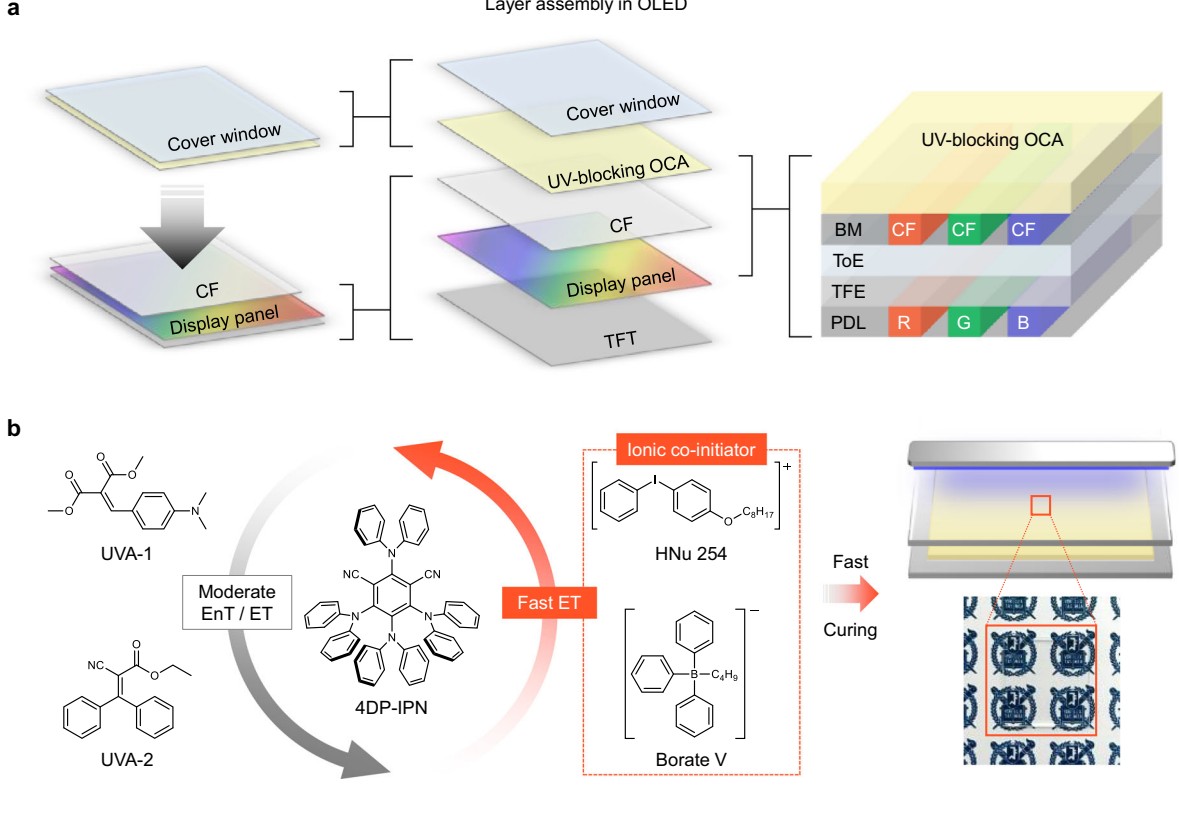

**Fig. 1 | Schematic illustration of this work. a** Device structure of a foldable display with UV-blocking optically clear adhesive (OCA) film; TFT, BM, CF, ToE, TFE, and PDL denote thin film transistor, black matrix, color filter, touch-panel on encapsulation, thin film encapsulation and pixel define layer, respectively. **b** Schematic illustration of this work; here EnT and ET denote triplet-triplet energy transfer and electron transfer, respectively. Image of the prepared UV-blocking OCA is shown. **c** Comparison of adhesive film curing rate without or with UV absorbers (UVA). The OCA films were cured by previous photo-initiating system (PIS) with tertiary amines (black line) and a newly designed PIS with HNu 254 and Borate V (orange line). Error bars represent the standard deviation of uncertainty in the reproducibility ($n = 5$).

A simple approach is to utilize visible light instead of UV light for curing, with the aid of a photosensitizer that efficiently absorbs visible light[14–32]; while visible-light curing can technically be used to add the UVAs to the CF layer, it is likely inefficient and thus impractical, mainly due to the high absorption of visible light by the pigments[12,13]. Kwon and colleagues have made noteworthy progress in the development of OCA for foldable displays in a recent study[33]. They achieved a substantial reduction in the amount of photocatalyst (PC) used in visible-light photocuring by employing a compound that generates long-lived lowest triplet excited states ($T_1$) as a PC[34–42]. This led to the successful production of UV-blocking OCA that exhibits high transparency in the visible-light range while also curing well under visible-light irradiation. However, a major drawback of this system was somehow the substantial reduction in curing speed when UVAs that do not absorb visible light is present (light dosage = 27,000 mJ m$^{-2}$ for 95.0% conversion), which falls short of the production speed required for its commercialization (Fig. 1c, black line in the plot on the right). This limitation ultimately affects the availability of UV-blocking OCAs on the market.

Here, we have developed a highly efficient PIS that operates under visible-light irradiation, allowing for the production of UV-blocking OCAs at a much faster rate compared to the previously reported PIS[33]. Through a mechanistic analysis of existing PIS, we found that electron transfer (ET) and triplet-triplet energy transfer (EnT) between the PC and the UVAs predominate over those between the PC and the co-initiator. This reduces the efficiency of previous PIS, consequently impeding the curing rate. To address this issue, we carefully replaced the existing PC (i.e., 4Cz-IPN) with 4DP-IPN, aiming to minimize ET and triplet-triplet EnT efficiency with UVAs (Fig. 1b). Additionally, we introduced co-initiators, i.e., HNu 254 and Borate V, that facilitate faster electron transfer and PC regeneration compared to previously used amine-based co-initiators. With such a PIS, we were able to produce UV-blocking OCAs at a rate fast enough for commercialization under 452 nm visible-light irradiation (light dosage = 2400 mJ cm$^{-2}$ for 98.1% conversion)[43], which is approximately 10 times faster than the previous system (Fig. 1c, orange line in the plot on the right). The resulting film exhibited excellent UV-blocking capabilities ($T_{345\,nm} \approx 1.3\%$), while maintaining high optical transparency ($T_{455\,nm} \approx 100.0\%$) and possessing viscoelastic properties suitable for use in foldable displays as OCA. Finally, to evaluate the practical UV-blocking performance of the prepared film, we conducted a UV-blocking test on an organic light-emitting diode (OLED) device. Compared to the device without the UV-blocking OCA, the OLED covered with our OCA film exhibited less reduction in luminance under UV irradiation ($\lambda_{max} = 368$ nm, 70 mW cm$^{-2}$), demonstrating its excellent UV-blocking capabilities. We believe that the UV-blocking OCA film developed in this study will find extensive practical applications, extending beyond the foldable OLEDs showcased, in providing protection against UV radiation for diverse smart devices with various form factors including epidermal[44–46], and wearable electronics[47,48].

## Results

### Origin of rate inhibition in the previous PIS
In the previous system, 4Cz-IPN and tertiary amine analogs (i.e., DMAEA and DMAEAc) were selected as a PC and co-initiators, respectively (Fig. 2a)[33]. Here, the amines are oxidized to transfer an electron to the $T_1$ state of the PC ($^3$PC$^*$), leading to the formation of α-amino radical species that act as initiators for the polymerization; 4Cz-IPN, a cyanoarene-based thermally activated delayed fluorescent (TADF) material, generates both lowest singlet ($S_1$) and $T_1$ excited states with negligible energy differences[35,36], and its $T_1$ concentration is approximately 100 times higher, predominantly contributing to ET (Supplementary Fig. 11). To obtain a clear understanding of the reduction in the curing rate resulting from UVAs in the previous

system, we conducted a Stern-Volmer relationship experiments by monitoring photoluminescence (PL) decay quenching on 4Cz-IPN, in the presence of each UVA (Fig. 2b and Supplementary Fig. 12). Dimethyl 2-(4-(dimethylamino)benzylidene)malonate (UVA-1) and ethyl 2-cyano-3,3-diphenylacrylate (UVA-2) were used to block the UVA (315–400 nm) and UV-B (280–315 nm) regions, respectively. The Stern-Volmer plots revealed that the quenching rate constants of 4Cz-IPN with UVAs ($k_q$ (UVA-1) = 3.2 × 10$^9$ M$^{-1}$ s$^{-1}$ and $k_q$ (UVA-2) = 7.5 × 10$^8$ M$^{-1}$ s$^{-1}$) are close to the diffusion limit, which is about 10–100 times faster than photoinduced ET (PET) with DMAEAc ($k_{PET}$ = 2.1 × 10$^7$ M$^{-1}$ s$^{-1}$) (Fig. 2c)[33]. These observations were consistent with the hindered curing rate of existing PIS in the presence of UVAs. To elucidate the origin of this dissipation of the PC in excited state, our initial hypothesis was that the triplet-triplet EnT played a role in the quenching phenomena. Based on our observations, the conditions for a Dexter-type triplet-triplet EnT[49], outlined below, were however not met for the case of PC and UVA-1, being i) $E_{PC}$ ($T_1$) > $E_{UVA}$ ($T_1$), ii) IP$_{PC}$− $E_{PC}$ ($T_1$) < EA$_{UVA}$, and iii) EA$_{UVA}$ + $E_{UVA}$ ($T_1$) < IP$_{PC}$, where IP and EA stands for ionization potential and electron affinity, respectively (Fig. 2d and Supplementary Fig. 15). In contrast, these criteria were satisfied in the case of UVA-2, which is in accordance with previous reports that UVA-2 acts as a triplet quencher[50,51]. In fact, the feasibility of PET between PC and UVAs can be assessed via the ground and excited state redox potentials of both PC and UVAs (Supplementary Table 3). Careful data analysis indeed suggested that UVA-1 is likely acting as reductant to reduce $^3$PC$^*$ ($E_{ox}^0$ = 0.95 V vs SCE) in the reductive quenching pathway (Fig. 2a). On the contrary, in the case of UVA-2, the less favorable driving force for PET suggests that the primary quenching process would likely be the triplet-triplet EnT process rather than ET. It is anyway noted that a quantitative analysis of the triplet-triplet EnT and ET processes involving UVAs is challenging, and further detailed analysis is currently conducted. To sum up, the main factors contributing to the inhibition in curing rate are thought to be the triplet-triplet EnT and ET between the $^3$PC$^*$ and the UVAs, which happen at a considerably faster rate compared to the ET between $^3$PC$^*$ and tertiary amines.

### Selection of a PC for the advanced PIS
We envisioned that by reducing the rates of triplet-triplet EnT and ET between $^3$PC$^*$ and UVAs, while simultaneously promoting ET between PC and co-initiators, the issue could be potentially resolved. To effectively suppress both triplet-triplet EnT and ET with UVAs concurrently, it becomes necessary to elevate the energy of highest occupied molecular orbital (HOMO) of PC, while maintaining the lowest unoccupied molecular orbital (LUMO) level less changed. This adjustment aims to reduce the driving force for the overall quenching processes involving UVAs. To resolve the issue, we screened the PC candidates established in our previous study, utilizing the systematic molecular catalyst design platform[36]. This platform enables systematic control over critical catalytic properties, such as redox potentials and the energies of $S_1$ and $T_1$, across a broad range. Among the PC candidates, 4DP-IPN emerged as an outstanding choice, primarily due to its higher HOMO energy, which results from substituting carbazole with the more electron-donating diphenylamine. In fact, the 4DP-IPN compound exhibited triplet energy of 2.38 eV and an excited state reduction potential ($E_{red}^*$ = 0.72 V), both approximately 0.30 eV and 0.75 eV lower than those of 4Cz-IPN, respectively (Fig. 3a). Interestingly, as compared to 4Cz-IPN, 4DP-IPN not only exhibited a longer triplet lifetime but also demonstrated superior $T_1$ generation (Fig. 3b)[33], which facilitate the reactive radical generation in the polymerization through the fast rate of ET process[52–55]. To fully leverage the excited-state reducing power of 4DP-IPN and maximize ET, we selected an electron-deficient co-initiator, an iodonium salt (i.e., HNu 254)[56–58]. According to the proposed mechanism, HNu 254 accepts electrons from $^3$PC$^*$, followed by the dissociation of the C−I bond, generating the required radicals for

**a** ■ Previously reported mechanism of the previous PIS

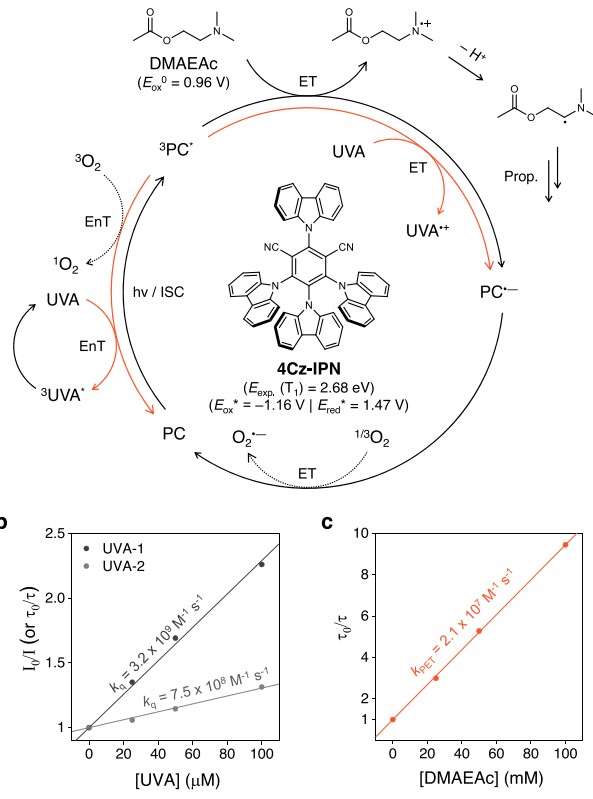

**a** ■ Proposed mechanism of this work

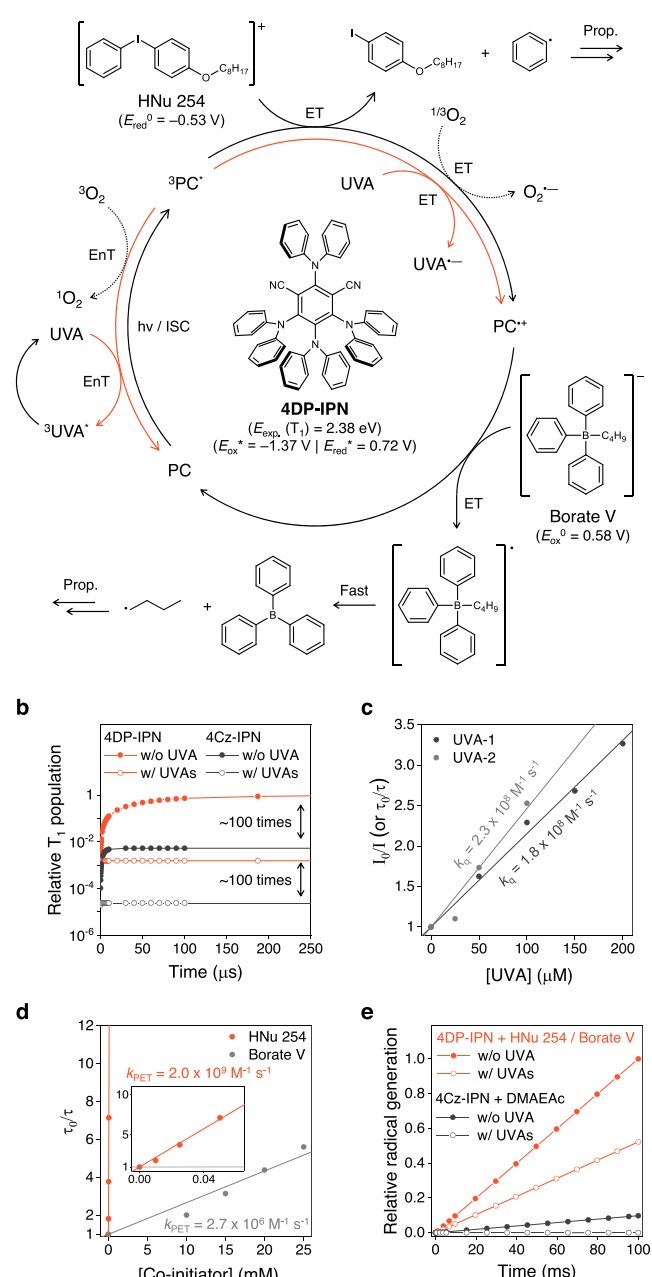

**d** ■ Electronic structure of PCs and UVAs obtained by experiments

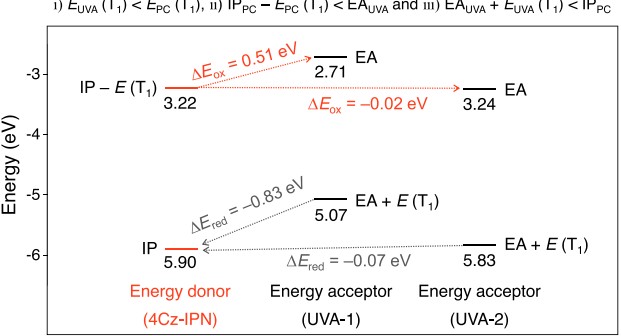

**Fig. 2 | Origin of inhibited curing rate in the previous PIS. a** Proposed mechanism of the previous PIS[33]. **b, c** Stern-Volmer plots for PL quenching of 4Cz-IPN in ethyl acetate along with addition of **b** UVAs and **c** DMAEAc at RT[33], respectively. Generally, PL quenching experiments were conducted by measurement of the change of PL decay with time-correlated single photon counting (TCSPC) techniques, but they were conducted with UVA-1 by monitoring the change in steady-state PL emission intensity maxima. **d** Electronic structure of PCs and UVAs obtained by experiments; here, IP and EA denote ionization potential and electron affinity, respectively (see Supplementary Table 2 for more detail). The driving forces (i.e., $\Delta E_{ox}$ and $\Delta E_{red}$) for each step of electron exchange in Dexter-type triplet-triplet EnT between 4Cz-IPN and UVAs are given; initial transfer pathway (orange dashed line) and following transfer pathway (gray dashed line).

**Fig. 3 | Newly designed PIS for UV-blocking OCA. a** Proposed mechanism of the newly designed PIS. **b** Kinetics simulation of relative $T_1$ population of 4DP-IPN (orange line) and 4Cz-IPN (black line) at 10 ppm in ethyl acetate without or with UVAs (UVA-1 (2000 ppm) and UVA-2 (6400 ppm)) under continuous 452 nm irradiation. **c, d** Stern-Volmer plots for PL quenching of 4DP-IPN in ethyl acetate along with addition of **c** UVAs and **d** HNu 254 and Borate V, respectively. Generally, PL quenching experiments were conducted by measurement of the change of PL decay with TCSPC techniques, but they were conducted with UVA-1 by monitoring the change in steady-state PL emission intensity maxima. **e** Kinetics simulation of radical generation in the newly designed PIS (orange line) and the previous PIS (black line) without or with UVAs mimicking homopolymerization of EHA under continuous 452 nm irradiation (see Supplementary Table 5 for more detail).

the initiation[59,60]. Moreover, we chose a borate salt (i.e., Borate V) as an electron-rich co-initiator[61,62]. This co-initiator facilitated the production of the radicals necessary for initiation while simultaneously promoting the rapid regeneration of the PC by donating electrons to the intermediate PC radical cation (PC[•+]).

## Evaluation of newly design PIS

To evaluate the effectiveness of the newly designed PIS, the ³PC* dissipation of 4DP-IPN was first monitored through PL quenching experiments with UVAs (Fig. 3c and Supplementary Fig. 13). A decrease

in the order of magnitude was observed for rate constants with UVAs ($k_q$ (UVA-1) = $1.8 \times 10^8\,M^{-1}\,s^{-1}$ and $k_q$ (UVA-2) = $2.3 \times 10^8\,M^{-1}\,s^{-1}$), as compared to those associated with 4Cz-IPN. We then conducted PL quenching experiments between 4DP-IPN and co-initiators. Based on the Stern-Volmer equation, we evaluated the rate constants as $k_{PET} = 2.0 \times 10^9\,M^{-1}\,s^{-1}$ for HNu 254 and $k_{PET} = 2.7 \times 10^6\,M^{-1}\,s^{-1}$ for Borate V (Fig. 3d), which correspond well with their redox potentials ($E_{red}^0$ (HNu 254) = $-0.53\,V$ and $E_{ox}^0$ (Borate V) = $0.58\,V$). Notably, the rate constant between 4DP-IPN and HNu 254 was found to be close to the diffusion limit, and approximately 1000 times faster than that of Borate V. These results suggest that our PIS relies primarily on an oxidative quenching cycle mediated by HNu 254, with Borate V being responsible for PC regeneration. In contrast to the previous system (4Cz-IPN and tertiary amines), the rate of ET is now competitive with the rate of $^3PC^*$ quenching by the UVAs, ensuring sufficient curing rates even in the presence of UVAs. Besides rapid ET, the process of radical generation by one-electron oxidized/reduced co-initiators, subsequent to ET, is critical for the curing rate. Upon ET, HNu 254 quickly decomposes into phenyl radicals through concerted ET[59,60]. Additionally, boranyl radicals generated from Borate V quickly produce alkyl radical species ($k_{diss}$ ~ $10^{11}\,s^{-1}$), rendering back electron transfer (BET) negligible[61,62]. However, amine-based co-initiators have a much slower rate constant ($k_{amine} = 2.3 \times 10^5\,M^{-1}\,s^{-1}$, Supplementary Fig. 18) for the generation of α-amino radical species after PET with PCs. Using the obtained rate constants, we performed simulations to estimate the amount of radical species generated by each PIS over time with the same amounts (i.e., 10 ppm) of PCs (Fig. 3e); as the quantitative analysis of the triplet-triplet EnT and ET processes with UVAs is challenging, to be simplified, triplet-triplet EnT with UVAs was considered as a primary quenching pathway of $^3PC^*$ without considering polymerization-factors such as radical propagation and termination. Strikingly, our results indicate that the newly designed PIS generates significantly more initiating radicals, approximately 1000 times more, particularly in the presence of UVAs. This should allow for fast and efficient curing to prepare UV-blocking OCAs.

## Preparation of UV-blocking OCAs

Based on the newly designed PIS, we prepared OCA films. Typically, acrylic OCA films are prepared using a two-step photocuring process (Supplementary information for the details)[63]. First, bulk polymerization of the monomer mixture is performed to obtain the appropriate viscosity for achieving the desired thickness of the final OCA film with high reproducibility. Then, the acrylic syrup, which has the desired viscosity, is cast to the desired thickness, leading to the formation of the ultimate OCA film through photocuring. Unlike conventional photoinitiator-based methods, our approach eliminates the need for additional PC incorporation in post-bulk polymerization. This is because the PC is regenerated through an ET process with a co-initiator in the catalytic cycle[33,64]. In our current experiments, we synthesized acrylic syrup through bulk polymerization of monomers under 455 nm irradiation. Subsequently, the resulting acrylic syrup was cast to a thickness of approximately 50 μm between two 100 μm-thick silicon-treated release films, followed by irradiation with 452 nm light for film curing.

We first reproduced OCAs using the same conditions as in our previous system, which involves the use of 4Cz-IPN and tertiary amines[29]. For this purpose, we prepared OCAs using the same combination of monomers, i.e., n-butyl acrylate (BA) and 4-hydroxybutyl acrylate (HBA) as indicated in Table 1 (entries 1 and 2) and Supplementary Table 6. Consequently, the OCA film required a light dosage of 5400 mJ cm$^{-2}$ to achieve a film conversion of over 95%. However, in the presence of UVAs, the light dosage required for film curing increased to 27,000 mJ cm$^{-2}$, which is approximately an order of magnitude higher than the light intensity needed for curing conventional OCAs[43]. Such a high light dosage is impractical for commercialization purposes.

Next, we prepared OCAs using HNu 254 and Borate V as co-initiators (500 ppm and 300 ppm with respect to monomers, respectively), while maintaining the same type and amount of PC (10 ppm of 4Cz-IPN with respect to monomers). Here, we replaced BA ($T_{g,\,poly(BA)} \approx -48\,°C$) with 2-ethylhexyl acrylate (EHA, $T_{g,\,poly(EHA)} \approx -58\,°C$) as the monomer[65]. This substitution offers the advantage of reducing the glass transition temperature ($T_g$) and providing significant flexibility even at RT. As a result, it imparts suitable adhesive strength and peel adhesion properties[66]. Furthermore, in this study, we optimized the process of incorporating UVAs. Previously, acrylic syrup was first prepared through bulk polymerization, followed by the addition of UVAs (0.3 and 1 phr (per hundred resin) for UVA-1 and UVA-2, respectively) and subsequent film formation through photocuring[67]. However, to enhance reproducibility, in this study, UVAs were added during the bulk polymerization step to produce the acrylic syrup, which was then cured to the film. As anticipated, the use of HNu 254 and Borate V exhibited significantly faster kinetics in film curing (light dosage = 3000 mJ cm$^{-2}$ for 94.3% conversion in Table 1, entry 3; see Supplementary Table 7 for comprehensive data on the optimization processes) compared to tertiary amines. However, the presence of UVAs inevitably reduced both

**Table 1 | Results of the visible-light-driven photo-initiating system to prepare the OCA films**

| Entry | PC | PC loading (ppm) | Co-initiator (ppm) | | UVA (ppm) | | Dosage $_{Film}$ (mJ cm$^{-2}$)[d] | Conversion $_{FT-IR}$ (%) | |
|---|---|---|---|---|---|---|---|---|---|
| | | | HNu 254 | Borate V | UVA-1 | UVA-2 | | Acrylic syrup | Film |
| 1[a] | 4Cz-IPN | 10 | DMAEAc (5000) | – | – | – | 5400 | 11.9 | 95.3 |
| 2[a] | 4Cz-IPN | 10 | DMAEAc (5000) | | 1500 (0.3 phr) | 4800 (1 phr) | 27,000 | 11.9 | 95.0 |
| 3 | 4Cz-IPN | 10 | 500 | 300 | – | – | 3000 | 10.3 | 94.3 |
| 4[b] | 4Cz-IPN | 10 | 500 | 300 | 2000 (0.3 phr) | 6400 (1 phr) | 6000 | 4.6 | 97.7 |
| 5 | 4DP-IPN | 1 | 500 | 300 | – | – | 3000 | 9.6 | 97.5 |
| 6[b] | 4DP-IPN | 1 | 500 | 300 | 2000 (0.3 phr) | 6400 (1 phr) | 6000 | 6.0 | 97.4 |
| 7 | 4DP-IPN | 2 | 1000 | 600 | – | – | 1800 | 19.3 | 98.5 |
| 8 | 4DP-IPN | 2 | 1000 | 600 | 2000 (0.3 phr) | 6400 (1 phr) | 3000 | 10.8 | 98.0 |
| 9[c] | 4DP-IPN | 3 | 1000 | 600 | – | – | 600 | 13.5 | 97.8 |
| 10 | 4DP-IPN | 3 | 1000 | 600 | 2000 (0.3 phr) | 6400 (1 phr) | 2400 | 8.3 | 97.8 |

OCAs were prepared with mixture of [EHA]:[HBA] = 3:1. For the synthesis of acrylic syrup, all reaction mixtures were degassed with $N_2$ for 30 min and then were irradiated by 455 nm irradiation (25 mW cm$^{-2}$) for 5 s at RT. The synthesized acrylic syrup was casted to a thickness of 50 μm between two 100 μm-thick silicon-treated release films, then it was cured to produce the OCA film under 452 nm irradiation (10 mW cm$^{-2}$). [a]Bulk polymerization was conducted for 30 s with [BA]:[HBA] = 4:1 under 455 nm irradiation (100 mW cm$^{-2}$), and then the film curing was conducted under 452 nm irradiation (15 mW cm$^{-2}$). [b,c]To obtain appropriately viscous acrylic syrup, bulk polymerizations were conducted for [b]10 s and [c]2 s, respectively. [d]Light dosage was evaluated from film curing time using the prepared acrylic syrup. All conversions were determined by the measurements of FT-IR spectroscopy.

bulk polymerization and curing processes, with film curing necessitating a substantial amount of light (light dosage = 6000 mJ cm$^{-2}$ in Table 1, entry 4).

Finally, we prepared OCAs using 4DP-IPN as the PC along with HNu 254 and Borate V as co-initiators (Table 1, entries 5–10; see Supplementary Tables 8–11 for comprehensive data regarding the optimization processes and reproducibility tests). By systematically varying the amount of PC and co-initiators, we identified the optimal conditions, achieving a catalyst reduction to 3 ppm—approximately three times lower than the previous PC, 4Cz-IPN (Table 1, entries 9 and 10). Under these optimized conditions, we achieved a conversion of 97.8% with a remarkably low light dosage of 600 mJ cm$^{-2}$. Even in the presence of UVAs, a conversion of 97.8% was attained with only 2400 mJ cm$^{-2}$ of light dosage, which is typically required integrated light intensity for conventional OCA film preparation[43]. These compelling results indicate the feasibility of mass-production of UV-blocking OCAs using this method. Furthermore, we made an intriguing observation within our system. We observed a strong dependency of monomer conversion on the light intensity employed. Higher irradiation intensity (i.e., 100 mW cm$^{-2}$) led to accelerated initial kinetics; however, the conversion reached a saturation point without further increase (Supplementary Fig. 21). This behavior can be attributed to the excessive generation of reactive radical species in the early stages, leading to accelerated termination between radical species. To address this challenge, we intentionally chose lower irradiation intensity for both bulk polymerization and film curing processes (i.e., 25 mW cm$^{-2}$ and 10 mW cm$^{-2}$, respectively), leading to improved monomer conversion. Under these conditions, we achieved a significant gel content in the resulting OCA films, without requiring additional crosslinkers (see Table 2). This outcome is likely attributed to the involvement of hydroxy groups in the crosslinking reactions between polymer chains in the presence of radical intermediates[66,68]. We are currently conducting an in-depth investigation into the detailed crosslinking mechanism involving the hydroxy group.

## Properties of the resulted OCAs

We tested the optical properties of the prepared OCA film and confirmed that it possesses UV-blocking capabilities comparable to commercial standards. To determine the film's transmittance, we employed a UV/vis spectrophotometer (Fig. 4c and Supplementary Fig. 23). The UV-blocking OCA exhibited remarkable effectiveness in blocking UV light ($T_{345\ nm} \approx 1.3\%$), while maintaining exceptional transparency in the visible-light range ($T_{455\ nm} \approx 100.0\%$). Given that the UV-blocking ability did not significantly differ across the OCA thickness range of 50–200 μm, we proceeded with a 50 μm thickness for subsequent experiments, aligning with the standard for foldable OCA films (see Supplementary Fig. 23a). We next evaluated the mechanical and adhesive properties of the prepared OCA film (Table 2 and Supplementary Figs. 25–28). As a control, a commercially available OCA for foldable displays (3 M, CEF 3602) was also tested. To verify the applicability as the OCA for foldable displays, we measured the viscoelastic properties through dynamic mechanical analysis (DMA)[69]. As illustrated in Table 2, entry 5, the UV-blocking OCA prepared in optimized conditions exhibits decent strain recovery, stress relaxation, and maximum stress characteristics that are comparable to those of CEF 3602.

We then proceeded to evaluate the peel strength of the prepared OCAs on both glass (as an alternative to ultrathin glass, UTG) and colorless polyimide (CPI), commonly used as cover windows in foldable displays (Table 2). The observed peel strength values are generally higher on the CPI than those on the glass, which would be attributed to the presence of phthalic anhydride derivatives[70] and benzidine derivatives[71] giving an abundant hydrogen bond site and the higher polarity to make interfacial interactions much stronger. In the case of the OCA without UVAs, the peel strength was similar to that of CEF 3602. However, a slight reduction in peel strength was observed in the UV-blocking OCA. This reduction may be attributed to UVAs with high concentrations migrating towards the surface of the acrylic adhesive film[72], or acting as defects that hinder adhesive properties[73]. Nevertheless, it is important to highlight that despite this slight decrease, the peel strength remains sufficient for the effective utilization of the OCA in foldable applications. To assess the high- and low-temperature reliability of the UV-blocking OCA film, we examined the viscoelastic behaviours of the OCA film across various temperatures[74]. The storage modulus ($G'$), loss modulus ($G''$), and damping factor (tan $\delta$) were measured by a rheometer through a dynamic temperature sweep ranging from −50 °C to 90 °C (Supplementary Table 12). Notably, the measured values at all temperatures met commercial OCA requirements. At low temperatures, the storage modulus value ($G' = 150.3$ kPa at −20 °C) was found to be slightly higher than that of CEF 3602 ($G' = 115.0$ kPa at −20 °C), yet it still meets the commercial standard where the storage modulus of commercially available OCAs is within the 100–150 kPa range[75,76]. Furthermore, the $T_g$ of the OCA film, measured at −39.7 °C during a dynamic temperature sweep, is consistent with the observed low $G'$ at −20 °C, as detailed in Supplementary Fig. 30. We also conducted dynamic folding tests to assess the suitability of the prepared UV-blocking OCAs for foldable displays (Supplementary Figs. 31, 32)[46,69]. The test specimen structure was designed based on an actual foldable smartphone with a fixed radius of curvature of 1.5 mm and a folding cycle of 0.5 Hz. Remarkably, the prepared OCAs demonstrated excellent folding durability within $5 \times 10^4$ folding cycles without cracks even at low temperatures[77], nearly meeting the specifications for foldable displays with similar values of surface texture (Δ$Z$) to CEF 3602 (Supplementary Fig. 32), thanks to their low storage modulus and high strain recovery at −20 °C (Supplementary Table 12).

**Table 2 | Properties of the prepared OCA film with different contents of UVAs**

| Entry | UVA (ppm) | | Gel contents (%) | Strain recover at 25 °C (%) | Stress relaxation at 25 °C (%)[b] | Max stress (kPa)[c] | Peel strength at 25 °C (N cm$^{-1}$) | |
|---|---|---|---|---|---|---|---|---|
| | UVA-1 | UVA-2 | | | | | Glass | CPI |
| 1 | CEF 3602 | | 68.4 (±1.0) | 86.1 (±2.2) | 47.5 (±1.3) | 43.1 (±6.1) | 4.8 (±0.1) | 3.7 (±0.2) |
| 2[a] | – | – | 76.8 (±1.0) | 84.0 (±1.5) | 54.6 (±1.2) | 64.1 (±7.0) | 4.8 (±0.4) | 5.3 (±0.2) |
| 3 | 2000 (0.3 phr) | – | 76.7 (±1.7) | 82.7 (±0.2) | 56.8 (±1.0) | 56.9 (±8.8) | 4.4 (±0.3) | 4.8 (±0.2) |
| 4 | – | 6400 (1 phr) | 79.5 (±2.0) | 81.1 (±1.5) | 55.1 (±0.7) | 53.2 (±12.1) | 3.8 (±0.3) | 4.1 (±0.4) |
| 5 | 2000 (0.3 phr) | 6400 (1 phr) | 77.2 (±1.0) | 84.7 (±2.2) | 54.2 (±2.5) | 64.0 (±6.9) | 3.1 (±0.2) | 4.0 (±0.1) |

OCAs were prepared with mixture of [EHA]:[HBA] = 3:1, 4DP-IPN (3 ppm), HNu 254 (1000 ppm), and Borate V (600 ppm). For the synthesis of acrylic syrup, all reaction mixtures were degassed with N$_2$ for 30 min and then were irradiated by 455 nm irradiation (25 mW cm$^{-2}$) for 5 s and [a]2 s at RT. The synthesized acrylic syrup was casted to a thickness of 50 μm between two 100 μm-thick silicon-treated release films, then they were cured under 452 nm irradiation (10 mW cm$^{-2}$). To assure fully cured OCA, all films were irradiated for 30 min. [b]Strain recovery was determined by the degree of recovery (%) over 5 min at RT after the specimen was kept for 10 min at 300% strain. [c]Max stress was evaluated as a peak stress value in the stress relaxation measurements at 300% strain.

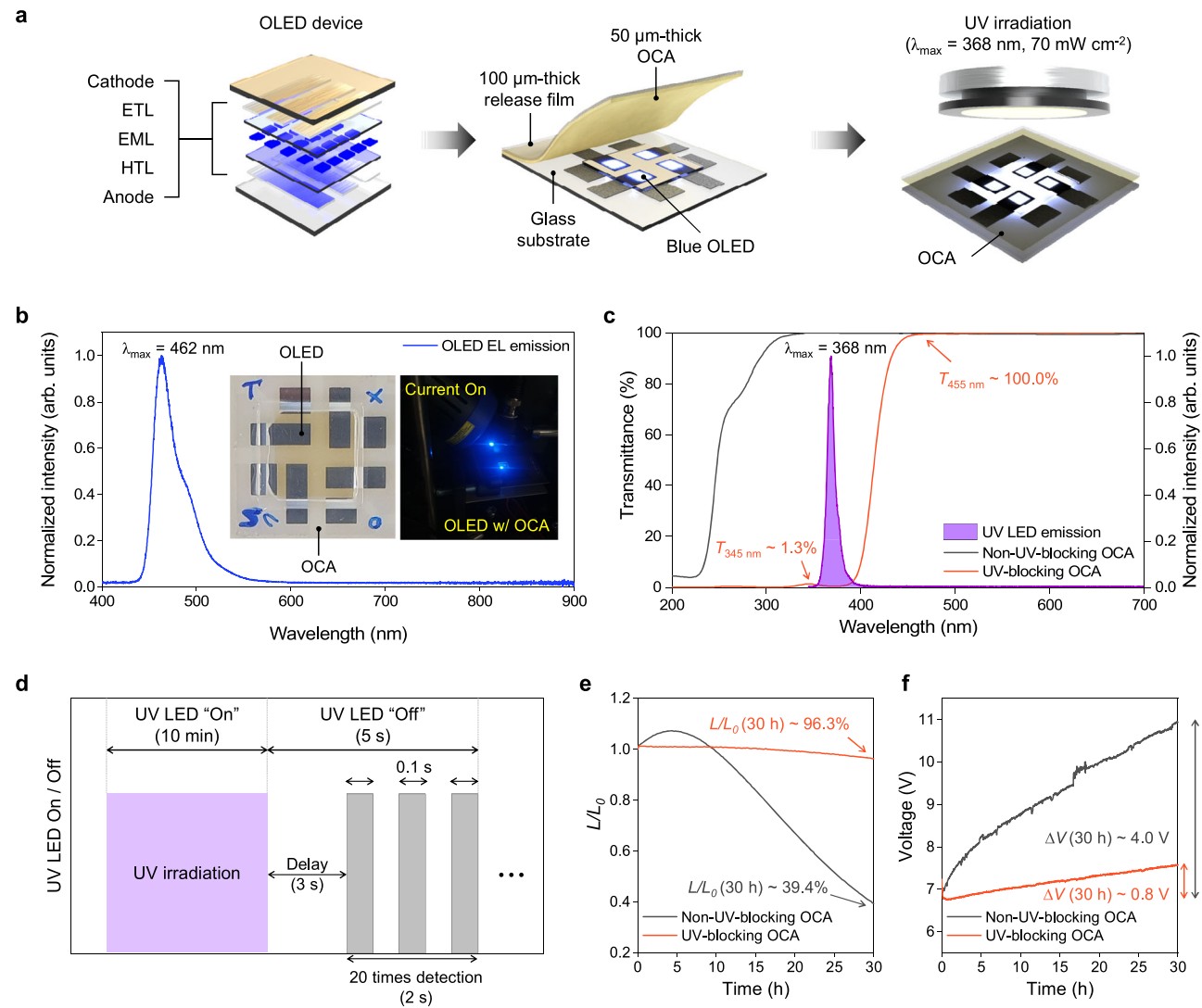

**Fig. 4 | Verification of UV-blocking OCA applied to OLED. a** Schematic illustration of a UV-blocking test; ETL, EML, and HTL denote electron transport layer, emission layer, and hole transport layer, respectively. **b** Electroluminescence (EL) emission spectra of tested OLED devices. Images of tested OLED devices and their EL emissions under the current ($J_0 = 100$ mA cm$^{-2}$) are shown (inset). **c** UV/vis transmittance of prepared OCAs and UV LED emission spectra. The transmittances of non-UV-blocking OCA film (black line) and UV-blocking OCA film (orange line) were measured with quartz plates as a cover glass. **d** Schematic illustration of the operation of UV-blocking test. During the measurement of luminance levels over time with a 10 min interval, the UV LED was momentarily deactivated for 5 s to prevent the detection of photoluminescent signals originating from the OLEDs due to UV irradiation. **e, f** Time-dependent changes of **e** luminance ($L_0 \sim 3690$ nit) and **f** voltage ($V_0 \sim 7.20$ V) of blue OLED covered by the prepared non-UV-blocking OCA film (black line) and UV-blocking OCA film (orange line) exposed by UV irradiation ($\lambda_{max} = 368$ nm, 70 mW cm$^{-2}$) under constant current ($J_0 = 100$ mA cm$^{-2}$). Changes in the luminance and voltage of blue OLED were recorded every 10 min.

## UV-blocking capability of the OCA

Finally, we assessed the UV-blocking capability of our OCA film when employed in blue fluorescent OLEDs (Fig. 4 and Supplementary Fig. 4). The choice of blue fluorescent OLEDs for experiments stemmed from the inherent vulnerability of their constituent molecules to UV irradiation, in contrast to their counterparts used in red and green OLEDs[78–80]. OLEDs covered by the OCA with and without UVA were operated at a constant current density of $J_0 = 100$ mA cm$^{-2}$ during continuous exposure to UV irradiation, exhibiting a peak wavelength of $\lambda_{max} = 368$ nm and a high intensity of 70 mW cm$^{-2}$. We compared the degradation in the optoelectronic performance of these devices with and without UV blockage. Ideally, for UV-blocking testing, all layers, including the CF layer, should be integrated to create an OLED device. However, due to practical difficulties in fabricating CF in the laboratory, we opted for a simplified device structure (Fig. 4a). Specifically, Fig. 4e, f show the temporal evolution of normalized luminance ($L$) with respect to its initial value ($L_0$) at $t = 0$, and the voltage rise (i.e., $\Delta V = V - V_0$) with $V_0$ as the initial voltage at $t = 0$, respectively (see Supplementary Fig. 24 for more detail). The UV-blocking OCA effectively shielded the blue OLED from UV exposure for 30 h, maintaining its normalized luminance ($L/L_0$) at 96.3%; On the other hand, the device covered by the non-blocking OCA experienced a significant reduction in its $L/L_0$ to 39.4%. The steeper voltage rise of $\Delta V = 4.0$ V vs 0.8 V for the non-UV-blocked and UV-blocked devices, respectively, indicates that the former device generated more degradation-induced defects due to stimulated molecular dissociation caused by UV exposure. We observed an unusual increase in the luminance during the early stages of UV irradiation, the underlying cause of which is currently under active investigation. The operational stability of OLEDs covered by the UVA-containing OCA clearly demonstrates the successful UV-blocking ability of our OCAs, even under harsh UV irradiation. This suggests that our UV-blocking OCA holds potential for the commercialization, boasting an extended product lifespan.

## Discussion

In summary, we have developed a highly efficient PIS that operates under visible-light irradiation. Through a thorough analysis of existing PIS, we identified the triplet-triplet EnT and ET with UVAs as the main factors limiting the curing rate. By employing the PC with higher HOMO energy and incorporating iodonium and borate salts as co-initiators with faster ET and radical generation, we addressed this issue. The resulting PIS allowed us to produce UV-blocking OCAs, achieving a high conversion (98.1%) with a light dosage of $2400 \, \text{mJ} \, \text{cm}^{-2}$, approximately 10 times lower than the previous system[33]. The lower light dosage and fast curing rate in our PIS highlight its economic potentials for the commercialization, making it comparable to the conventional UV-curing process[43]. The prepared OCAs exhibited excellent UV-blocking capabilities while maintaining high optical transparency and demonstrated fascinating physico-chemical properties, including adhesive properties, viscoelasticity, and folding durability, comparable to those of the commercial OCA. Moreover, we successfully demonstrated their practical UV-blocking capabilities by applying them to OLED devices, indicating their potential for various applications that require UV-blocking abilities. We believe that the UV-blocking OCA prepared here will have enormous practical applications beyond the foldable OLEDs, providing protection against UV radiation for diverse smart devices with various form factors. Furthermore, the newly developed PIS is believed to be used as a key PIS not only for the preparation of (functional) OCA film, but also visible-light-based 3D/4D printing[81–85], optically clear resins[33], soft actuator/robotics[86,87], dental resin[88,89], and more[90].

## Methods

### General experimental procedures for synthesis of acrylic syrup

A 20 ml vial (glass, Sungho SIGMA) equipped with a stirring bar was charged with monomer mixture, PCs, and co-initiators. Afterwards, the vial was capped with a rubber septum sealed with parafilm and degassed with 99.999% $N_2$ for 30 min. PCs and co-initiators were used after dilution in acrylate monomers for reproducibility. In the case of addition of UVAs, they were added in this reaction solution before the bulk polymerization in newly designed PIS. During the degassing process, the reaction solution was kept in a dark condition to block the room light preventing undesired polymerization. Subsequently, acrylic syrup was synthesized under irradiation of 3 W MR 16 LED ($\lambda_{max} = 455 \, \text{nm}$, $25$–$100 \, \text{mW} \, \text{cm}^{-2}$) at RT.

### General experimental procedures for film curing

For film curing, two 15 W string-type blue LEDs ($\lambda_{max} = 452 \, \text{nm}$) were used, and their intensities were set up as $10 \, \text{mW} \, \text{cm}^{-2}$. Film curing was carried out by additionally irradiating visible light to the prepared acrylic syrup. Since the PIS used in this process reuses the PC within the catalyst cycle of the acrylic syrup preparation process, no further additives were added during the film process. Moreover, a crosslinking agent was also not included in this process. For film curing, acrylic syrup was coated between two release films and casted to a thickness of $50 \, \mu\text{m}$ using a film casting applicator. The samples for measurements of optical and viscoelastic properties were cured for 40 min for the previous PIS (i.e., 4Cz-IPN and tertiary amines) and 30 min for this PIS (i.e., 4DP-IPN, HNu 254 and Borate V) to obtain fully cured OCAs.

### General experimental procedures for UV-blocking test of OLEDs

Both an OLED covered by non-UV-blocking OCA and the same device with UV-blocking OCA underwent simultaneous illumination using a UV LED source (Kessil, PR160L) with a peak wavelength of $\lambda_{max} = 368 \, \text{nm}$ and an intensity of $70 \, \text{mW} \, \text{cm}^{-2}$ at RT. Both devices were consistently operated under a constant current density of $J = 25$–$100 \, \text{mA} \, \text{cm}^{-2}$, utilizing a multi-channel source measure unit (Agilent Technologies, U2722A). During the measurement of

luminance levels over time with a 10 min interval, we momentarily deactivated the UV LED for 5 s to prevent the detection of photo-luminescent signals originating from the OLEDs due to UV irradiation. Upon turning off the UV LED, a waiting period of 3 s was included, followed by a 2 s measurement of electroluminescence from the OLEDs. This measurement involved the averaging of 20 collected signals, each spaced at 0.1 s intervals, in order to minimize detection errors. Degradation of OLEDs' luminance over time was captured using a photodiode (Thorlabs, SM1PD1A) and subsequently recorded through a data logger (Keysight, DAQ970A). All stages of OLED testing were fully automated through an in-house LabVIEW virtual instruments.

## Data availability

The authors declare that the data supporting the findings of this study are available within the paper and its Supplementary Information.

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

## Acknowledgements

This work was supported by the Technology Innovation Program (20011317, Development of an adhesive material capable of morphing more than 50% for flexible devices with a radius of curvature of 1 mm or less) and by the Industrial Strategic Technology Development Program (20011059, Development of inks for emitting layers with high performance and long lifetime), which were funded by the Ministry of Trade, Industry & Energy (MOTIE, Korea). It was also backed by the National Research Foundation of Korea (NRF) grants provided by the Korean government (MSIT) under grant numbers 2021R1A5A1030054 and NRF-2022R1A2C2011627. The work in Madrid was supported by the Spanish Ministerio de Ciencia, Innovación y Universidades, and the European Structural Funds through projects PID2022-138222NB-C21, CEX2020-001039-S, and by the Campus of International Excellence (CEI) UAM + CSIC.

## Author contributions

Y.Kwon, S.L., and M.S.K. were responsible for the initial conception of the project and wrote the initial draft of the manuscript. Y.Kwon and W.J. performed the photophysical measurements and CV measurements. S.L., J.K., and Y.P. synthesized the OCA films and measured the mechanical- and optical properties of OCA films. J.J. fabricated OLED devices and performed the UV-blocking test supported by S.L and J.K. under supervision from J.L. S.L., Y.P., Y.Kim, and H.J.K. discussed the mechanical properties of OCA films. Y.Kwon, J.G., and M.S.K. were involved in the discussion of the photophysics and photochemical reactions. Y.Kwon performed DFT calculations and the kinetic simulation under supervision from M.S.K. All the authors discussed the results and commented on the manuscript. J.L., Y.Kim, and M.S.K. supervised the project and were responsible for editing the final manuscript.

## Competing interests

The patent application (K.R. application number: 1020230026669 and 1020230089201) was filed by M.S.K., Y.Kwon, and S.L. from Seoul National University R&DB Foundation. The patent application covered the preparation of methods for preparing photocured resin and their application into UV-blocking adhesive. The remaining authors declare no competing interests.
