## [Peer Review File · Nature Communications]

Ultraviolet Light Blocking Optically Clear Adhesives for Foldable Displays via Highly Efficient Visible-Light CuringEditorial Note: Parts of this Peer Review File have been redacted as indicated to maintain the confidentiality of unpublished data.

REVIEWER COMMENTS

Reviewer #1 (Remarks to the Author):

In their work, the authors greatly improved the visible-light curing speed by developing a highly efficient photo-initiating system, based upon their previous research work on the synthesis of ultraviolet light blocking OCA. The key mechanical properties of OCA that are utmost important to the structural integrity of OCA bonded foldable screens, such as adhesion strength, stress relaxation, and strain recovery, have been analyzed and showed good results comparable to traditional commercial OCAs. This work presents an important progress in developing UV-blocking soft adhesives for foldable screens.

The manuscript is well written. Below are some minor questions and a few typos.

1. What are the advantages of the stress relaxation followed by strain recovery control mode used in this work, compared to the classical separated creep recovery and stress relaxation tests? In the stress relaxation step, is it instantaneous loading or ramped loading being used to apply the strain to 300%? What is the strain rate if it is ramped loading?
2. In the rheology test, the temperature sweep test in the range of -20 °C to 80 °C has been conducted, can the glass transition temperature (T_g) of the material be identified from the data?
3. For the peeling test, what is the major debonding failure mode, cohesive failure or adhesive failure? Can the authors provide experiment pictures showing the peeling process?
4. In page 6, the authors wrote "Notably, the measured values at all temperatures met commercial OCA requirements. ... , still fulfilling the standard for commercialization ", are there any specific numerical ranges in the commercial OCA requirements? Can the authors provide the references that describe the specific commercial requirements or standards?
5. In Table 2 caption, "contains" can be replaced with "contents"
6. In abstract, "(PIS)" was defined twice.

Reviewer #2 (Remarks to the Author):

To address the need for reducing power consumption in foldable smartphones, the development of an advanced optically clear adhesive (OCA) capable of blocking UV light while preserving crucial properties like adhesion and strain relief is imperative. The authors introduce a novel photo-initiating system (PIS) that enables UV-blocking OCA production at a rate 10 times faster than their previous work when exposed to visible-light irradiation. While the overall design is interesting, the manuscript's presentation lacks clarity in conveying the innovations, potentially leading to reader misinterpretation.

My comments are outlined below:

1. In the Abstract and Introduction, the authors assert that they employed a newly designed PC (PhotoCatalyst), specifically 4DP-IPN. However, it is worth noting that this PC, 4DP-IPN, has been previously explored in their earlier research. Throughout the manuscript, substantial effort is invested in comparing 4DP-IPN with 4Cz-IPN, both in the main text and Figure 1. This extensive comparison may inadvertently obscure the innovative aspects and significance of the current study. It is essential to emphasize that the primary distinction between the photo-initiating systems (PIS) developed in this work and their prior research lies in the choice of the photoinitiator. This pivotal distinction could be highlighted more prominently in the present manuscript. Additionally, Figures 1 and 2 contain a large amount of redundant information from their earlier work, warranting attention for a more concise and focused presentation.
2. The molecules are initially excited to the singlet state and subsequently transition into the triplet state in the photoexcitation. The emphasis in this work primarily centers on the design principles pertaining to the triplet state of various species within the photo-initiating system (PIS). However, the question arises: why is the singlet state of these materials not equally critical and deserving of discussion?
3. It's better to differentiate more for the terms "triplet-triplet energy transfer (EnT)" and "energy transfer (ET)".
4. The authors achieved developing UV-blocking OCAs at a rate approximately 10 times faster than before. And claimed that the technique could be immediately commercialized. However, the manuscript's current presentation may leave readers seeking more compelling evidence to fully substantiate this claim.
5. What about the UV blocking ability of the layers "BM and CF", "ToE", and "TFE"? In the dome of Fig. 4, it might be better to include those layers for the comparison.

6. For the adhesive test, please provide the testing conditions and the related standard. The peel strength in the newly proposed OCA is lower than in their previous work, what could be the potential reasons?

7. What advantages does the design of the OCA with UV-blocking ability offer over alternative strategies, like incorporating UVAs into the BM-CF layer? This alternative approach might potentially expedite OCA development significantly, given its potentially reduced demands on the choice of PC and initiators.

8. How does the UV-blocking ability depend on the thickness of the OCA?

9. The misspelling of 'TFE' is found in the Fig.1 legend.

Reviewers' comments to author:

We are grateful to the reviewers for their constructive comments and suggestions, which greatly helped to clarify a number of important points, avoid misunderstandings, and improve the clarity and referencing of the paper. The reports of the reviewers were overall very positive on the significance and novelty, stressing the suitability of our work for publication in *Nature Communications*. We have carefully revised our manuscript in line with all the reviewers' comments. The point-by-point response is given below. Our responses to reviewers' comments are highlighted in **blue color** with a highlight in the revised manuscript and SI.

Reviewer #1 (Remarks to the Author):

In their work, the authors greatly improved the visible-light curing speed by developing a highly efficient photo-initiating system, based upon their previous research work on the synthesis of ultraviolet light blocking OCA. The key mechanical properties of OCA that are utmost important to the structural integrity of OCA bonded foldable screens, such as adhesion strength, stress relaxation, and strain recovery, have been analyzed and showed good results comparable to traditional commercial OCAs. This work presents an important progress in developing UV-blocking soft adhesives for foldable screens.

The manuscript is well written. Below are some minor questions and a few typos.

1) What are the advantages of the stress relaxation followed by strain recovery control mode used in this work, compared to the classical separated creep recovery and stress relaxation tests? In the stress relaxation step, is it instantaneous loading or ramped loading being used to apply the strain to 300%? What is the strain rate if it is ramped loading?

Response: Thank you for the reviewer's careful comments. It appears there was some confusion caused by our insufficient explanation, for which we apologize. To clarify, we measured the strain recovery and stress relaxation of our OCAs separately. Initially, we determined the stress corresponding to a 300% strain from dynamic mechanical analysis. Subsequently, we used this stress level to measure strain recovery.

We also wish to highlight that our focus was on the measurement of strain recovery rather than creep recovery, given our targeted applications in foldable displays. Creep recovery typically involves long-term deformation under constant stress until the material reaches a viscous deformation region. The reviewer correctly noted that foldable durability of an OCA can be evaluated by both creep recovery and stress relaxation, as outlined in '*Pressure-Sensitive Adhesives for Flexible Display Applications* (2019)'. However, considering the real-world use of foldable displays, an OCA must exhibit durability during rapid folding and unfolding processes, and it needs to maintain its integrity when the display is kept in a folded state for extended periods.

Fig. R1 Schematic illustration of 180° folded OCA in the foldable display.

To simulate the practical use of foldable displays, we conducted the strain recovery tests by applying instantaneous loading at a constant strain, confined to the elastic deformation region. For additional context, applying a 300% strain is a common standard for testing the strain recovery of OCAs used in foldable displays. This level of strain correlates with the curvature that these displays typically experience in practical applications (see **Fig. R1**; now **Supplementary Fig. 20** in the revised SI). For instance, the shear behavior of an OCA can be illustrated by the difference in length between an inner curve (**Fig. R1**, blue line) and an outer curve (**Fig. R1**, green line), as represented by **Equations 1–3**.

$$D_1 = (a + T) \times \pi \quad (1)$$

$$D_2 = a \times \pi \quad (2)$$

$$\text{Shear strain} = \frac{D}{T} = \frac{D_1 - D_2}{T} = \frac{T \times \pi}{T} = \pi \quad (3)$$

In the given context, D_1 and D_2 represent the outer and inner folding curves, respectively, while D signifies a displacement in the shear strain. T corresponds to the thickness of the OCA, and a is the folding radius. It follows from the relationship in the shear strain that the displacement is likely to increase in proportion to the thickness, multiplied by π . Consequently, under the conditions of 180° folding, we applied a strain of approximately 300% in order to assess the strain recovery and stress relaxation.

2) In the rheology test, the temperature sweep test in the range of -20 °C to 80 °C has been conducted, can the glass transition temperature (T_g) of the material be identified from the data?

Response: Thank you for the reviewer's insightful comment. Because the T_g of the OCA we prepared is beyond the temperature range covered by the original manuscript's temperature sweep test, it was not observed in the initial data. In response to the reviewer's comments, we expanded the temperature range from -50°C to 90°C and conducted another temperature sweep. The T_g was measured to be approximately -40°C, as illustrated in **Fig. R2** below (now **Supplementary Fig. 21** in the revised SI).

Fig. R2 The damping factor ($\tan \delta$) of the prepared OCA film was measured using a rheometer with a dynamic temperature sweep from -50°C to 90°C. For the prepared OCAs with a [EHA]:[HBA] ratio of 3:1, we used 4DP-IPN (3 ppm), HNu 254 (1000 ppm), and Borate V (600 ppm). Additionally, to create a UV-blocking OCA, we incorporated UVA-1 (2000 ppm) and UVA-2 (6400 ppm).

Generally, the T_g of OCAs used in foldable displays is targeted to be below -20°C to ensure adequate flexibility (*SID Symposium Digest of Technical Papers*, **48**, 198 (2017)). This target also takes into account the use of the display in extremely cold environments (*Samsung Display Newsroom. Samsung Foldable OLED: 'Folding Test in Extreme Cold'*). Therefore, in synthesizing OCAs, we prepared UV-blocking OCAs using 2-ethylhexyl acrylate (EHA, T_g (poly(EHA)) ~ -58°C) and 4-hydroxybutyl acrylate (HBA, T_g (poly(HBA)) ~ -65°C) (*BASF, Acrylic and Methacrylic Monomers* (2021)). With the combination of these acrylic monomers, we anticipated, based on the Flory-Fox equation (**Equation 4**), that the T_g of the prepared OCA would theoretically fall within the range of -65 to -58°C; in this equation, w_x represents the weight fraction of component x, and $T_{g,x}$ denotes the glass transition temperature of component x (where x = A or B).

$$\frac{1}{T_g} = \frac{w_A}{T_{g,A}} + \frac{w_B}{T_{g,B}} \quad (4)$$

Despite expectations based on a [EHA]:[HBA] ratio of 3:1 (i.e., a weight percentage ratio of EHA:HBA = 3.83:1), the actual T_g of our synthesized OCAs was observed to be -39.7°C from a dynamic temperature sweep ranging from -50°C to 90°C (**Fig. R2**). We have properly added these results in the revised manuscript as follows:

“The storage modulus (G'), loss modulus (G''), and damping factor ($\tan \delta$) were measured by a rheometer through a dynamic temperature sweep ranging from $-50\text{ }^{\circ}\text{C}$ to $90\text{ }^{\circ}\text{C}$ (*Supplementary Table 12*). Notably, the measured values at all temperatures met commercial OCA requirements. At low temperatures, the storage modulus value ($G' = 150.3\text{ kPa}$ at $-20\text{ }^{\circ}\text{C}$) was found to be slightly higher than that of CEF 3602 ($G' = 115.0\text{ kPa}$ at $-20\text{ }^{\circ}\text{C}$), yet it still meets the commercial standard where the storage modulus of commercially available OCAs is within the 100–150 kPa range.^{70,71} Furthermore, the T_g of the OCA film, measured at $-39.7\text{ }^{\circ}\text{C}$ during a dynamic temperature sweep, is consistent with the observed low G' at $-20\text{ }^{\circ}\text{C}$, as detailed in *Supplementary Fig. 21*.”

3) For the peeling test, what is the major debonding failure mode, cohesive failure or adhesive failure? Can the authors provide experiment pictures showing the peeling process?

Response: We appreciate the referee’s thorough comments. Generally, the failure modes in peeling tests are categorized as adhesive failure, cohesive failure, and structural failure (*Adhesives Technology Handbook (Second Edition)*, 1–19 (2009)). Adhesive failure typically occurs where there is insufficient physical bonding strength between the adhesive and the adherend, resulting in sticky-slip. In contrast, cohesive failure originates from a breakdown of the intermolecular bonding strength (i.e., cohesive force) within the adhesive itself. Structural failure refers to the failure of the entire structure, which implies an unbreakably strong bond and adhesion. In light of these failure modes, we observed no residue on the substrate during the peeling tests conducted in this work, indicating an absence of failure.

Fig. R3 Images show a representative peel strength test conducted on a glass substrate. To affix the specimens (OCAs) onto the glass substrate secured in the clamp, we used a commercially available acrylic adhesive, 3M Scotch™ Tape 810. The peel strength of the prepared OCAs was measured after 24 hours of attachment.

However, in response to comment #3—which hypothesized that failures occurred in the peel strength test—we surmise that adhesive failure would be the most likely. Although the common monomer ratio between EHA and HBA is 4:1 for synthesizing foldable OCAs (*Patent US 10640689 B2*), we utilized a ratio of $[\text{EHA}]:[\text{HBA}] = 3:1$. This was done to enhance the crosslinking in our prepared OCAs, thereby preventing cohesive failure (*Int. J. Adhes. Adhes.*, **74**, 137–143 (2017); *Prog. Org. Coat.*, **88**, 155–163 (2015); *Polym. Int.*, **52**, 347–357 (2003)). Furthermore, to provide a clearer understanding of our process, we have included images of the peeling test in the revised SI (*Fig. R3*).

4) In page 6, the authors wrote "Notably, the measured values at all temperatures met commercial OCA requirements. ... , still fulfilling the standard for commercialization ", are there any specific numerical ranges in the commercial OCA requirements? Can the authors provide the references that describe the specific commercial requirements or standards?

Response: Thank you for the reviewer's insightful comments. Unlike standardized methods for measuring physical properties, universally accepted standards for the commercialization of these properties do not exist; they vary among companies and are often closely guarded as trade secrets. For example, the standards for the physical properties of flexible OCA differ between companies such as LG Display and Samsung Display. Consequently, it is difficult to establish precise benchmarks for commercialization; please be aware that disclosing this information in your paper could potentially lead to legal issues. In the case of the OCA developed in this study, it has been internally evaluated by Samsung Display and found to nearly meet their commercialization standards. Additionally, we can draw on the approximate commercial standards based on 3M's CEF 3602, which is already being utilized in foldable devices.

Therefore, to assess the commercial viability of our OCA films in this study, we compared their viscoelastic properties with those of the commercially available foldable OCA, specifically 3M's CEF 3602. **Table R2** illustrates that the UV-blocking OCA produced under our conditions demonstrates a storage modulus (G'), loss modulus (G''), and damping factor ($\tan \delta$) comparable to those of CEF 3602, both at room temperature and at elevated temperatures. At lower temperatures ($-20\text{ }^{\circ}\text{C}$), our OCA exhibits slightly higher storage modulus values than CEF 3602. However, considering that the storage modulus for commercialized OCAs ranges from 100 to 150 kPa at $-20\text{ }^{\circ}\text{C}$, as reported in the *SID Symposium Digest of Technical Papers*, **48**, 198 (2017) and *Patent US 11502270 B2*, our OCA—with a G' of 150.3 kPa—demonstrates strong potential for commercialization. We have included these references in the revised manuscript to further substantiate our claims.

Table. R2 Results for the storage modulus (G'), loss modulus (G''), and damping factor ($\tan \delta$) of the OCA films prepared in this study are presented. The viscoelastic properties of these OCAs were evaluated at $-20\text{ }^{\circ}\text{C}$, $25\text{ }^{\circ}\text{C}$, $60\text{ }^{\circ}\text{C}$, and $85\text{ }^{\circ}\text{C}$. For comparison, CEF3602 was used as a benchmark in the control experiments.

Entry	Specimen	Storage modulus (G') (kPa)				Loss modulus (G'') (kPa)				Damping factor ($\tan \delta$)			
		$-20\text{ }^{\circ}\text{C}$	$25\text{ }^{\circ}\text{C}$	$60\text{ }^{\circ}\text{C}$	$85\text{ }^{\circ}\text{C}$	$-20\text{ }^{\circ}\text{C}$	$25\text{ }^{\circ}\text{C}$	$60\text{ }^{\circ}\text{C}$	$85\text{ }^{\circ}\text{C}$	$-20\text{ }^{\circ}\text{C}$	$25\text{ }^{\circ}\text{C}$	$60\text{ }^{\circ}\text{C}$	$85\text{ }^{\circ}\text{C}$
1	CEF 3602	115.0	45.3	32.3	27.6	78.8	12.4	10.0	8.7	0.69	0.27	0.31	0.32
2	Our OCA	150.3	45.0	36.1	32.7	145.4	11.1	8.8	8.1	0.97	0.25	0.24	0.25

5) In Table 2 caption, "contains" can be replaced with "contents"

Response: Following the reviewer's comments, we have properly corrected.

6) In abstract, "(PIS)" was defined twice.

Response: Following the reviewer's comments, we have properly corrected.

Reviewer #2 (Remarks to the Author):

To address the need for reducing power consumption in foldable smartphones, the development of an advanced optically clear adhesive (OCA) capable of blocking UV light while preserving crucial properties like adhesion and strain relief is imperative. The authors introduce a novel photo-initiating system (PIS) that enables UV-blocking OCA production at a rate 10 times faster than their previous work when exposed to visible-light irradiation. While the overall design is interesting, the manuscript's presentation lacks clarity in conveying the innovations, potentially leading to reader misinterpretation.

My comments are outlined below:

1) In the Abstract and Introduction, the authors assert that they employed a newly designed PC (Photocatalyst), specifically 4DP-IPN. However, it is worth noting that this PC, 4DP-IPN, has been previously explored in their earlier research. Throughout the manuscript, substantial effort is invested in comparing 4DP-IPN with 4Cz-IPN, both in the main text and Figure 1. This extensive comparison may inadvertently obscure the innovative aspects and significance of the current study. It is essential to emphasize that the primary distinction between the photo-initiating systems (PIS) developed in this work and their prior research lies in the choice of the photoinitiator. This pivotal distinction could be highlighted more prominently in the present manuscript. Additionally, Figures 1 and 2 contain a large amount of redundant information from their earlier work, warranting attention for a more concise and focused presentation.

Response: We thank the reviewer for the encouraging comments and constructive suggestions. We acknowledge the reviewer's point that an excessive focus on the previous work may have detracted from the uniqueness of this study, and the detailed characterizations of the chemicals could have obscured the clarity in *Fig. 1* and *2*. In response to the referee's remarks, we thus have substantially revised the abstract and introduction to more clearly delineate the differences between our previous and current work (see below). Additionally, *Fig. 1* and *2* have been updated to enhance the readers' comprehension of our novel contributions. Based on this, we clarified the issues present in the past studies to better convey the novelty of our current research (see *Fig. R4*; *Fig. 1b* in the revised manuscript). Moreover, as suggested by the reviewer, we have incorporated references to previous studies that used 4DP-IPN, offering a more comprehensive understanding of its application.

The abstract of the revised manuscript has been rewritten as follows:

"In developing an organic light-emitting diode (OLED) panel for a foldable smartphone (specifically, a color filter on encapsulation) aimed at reducing power consumption, the use of a new optically clear adhesive (OCA) that blocks UV light was crucial. However, the incorporation of a UV-blocking agent within the OCA presented a challenge, as it restricted the traditional UV-curing methods commonly used in the manufacturing process. Although a visible-light curing technique for producing UV-blocking OCA was proposed, its significantly slow curing speed posed a barrier to commercialization. Our study introduces an innovative and efficient photo-initiating system (PIS) for the rapid production of UV-blocking OCAs utilizing visible light. We have re-engineered the photocatalyst (PC) to minimize electron and energy transfer to UV-blocking agents and introduced new co-initiators that allow for faster electron transfer and quicker PC regeneration compared to previously established amine-based co-initiators. This advancement enabled a tenfold increase in the production speed of UV-blocking OCAs, while maintaining their essential protective, transparent, and flexible properties. When applied to OLED devices, this new OCA demonstrated outstanding UV

protection, suggesting its potential for broader application in the safeguarding of various smart devices.”

The following sentences have now been added in the introduction:

“Through a mechanistic analysis of existing PIS, we found that electron transfer (ET) and energy transfer (EnT) between the PC and the UVAs predominates over that between the PC and the co-initiator. This greatly reduces the efficiency of PIS and, consequently, impedes the curing rate. To address this issue, we delicately modified the structure of the existing PC to design a new PC, 4DP-IPN, with reduced ET and EnT efficiency with UVAs (**Fig. 1b**). Additionally, we introduced co-initiators, i.e., HNu 254 and Borate V, that facilitate faster electron transfer and PC regeneration compared to previously used amine-based co-initiators.”

Fig. R4 Schematic illustration of this work. Schematic illustration of this work; here EnT and ET denote energy transfer and electron transfer, respectively. Image of the prepared UV-blocking OCA is shown.

2) The molecules are initially excited to the singlet state and subsequently transition into the triplet state in the photoexcitation. The emphasis in this work primarily centers on the design principles pertaining to the triplet state of various species within the photo-initiating system (PIS). However, the question arises: why is the singlet state of these materials not equally critical and deserving of discussion?

Response: We appreciate the reviewer's insightful comments, which have certainly helped to prevent confusion among potential readers. We apologize for any misunderstanding caused by our previous insufficient explanation.

a ■ Measurements of photoinduced electron transfer rate constant (k_{PET} , $M^{-1}\cdot s^{-1}$) between PC and co-initiators

b ■ Scheme of kinetic simulation of PC concentration in excited state under photostationary state

Fig. R5 Comparison of contributions to electron transfer from the singlet and triplet excited states of 4DP-IPN and 4Cz-IPN. (a) The photoinduced electron transfer rate constants (k_{PET} , $M^{-1}\cdot s^{-1}$) between the PCs and co-initiators were measured. These constants for both the singlet and triplet excited states of the PCs were determined by monitoring the changes in prompt fluorescence (PF) and delayed fluorescence (DF) using time-correlated single-photon counting (TCSPC) techniques at an excitation wavelength (λ_{ex}) of 377 nm and a detection wavelength (λ_{det}) of 520 nm. The photoluminescence (PL) decay spectra at room temperature were obtained from degassed solutions of PCs in ethyl acetate (1.0×10^{-5} M) with varying concentrations of quenchers. (b) The simulated relative populations of the lowest singlet (S_1) and triplet (T_1) excited states of the PCs in ethyl acetate (4.75×10^{-5} M) were calculated under the photostationary state during illumination with a 452 nm LED at an intensity of $10 \text{ mW}\cdot\text{cm}^{-2}$.

As the reviewer has pointed out, cyanoarenes, a class of thermally activated delayed fluorescence (TADF) compounds, can exhibit both excited singlet and triplet states (*Chem. Soc. Rev.*, **50**, 7587 (2021)). While both singlet and triplet states in 4DP-IPN and 4Cz-IPN demonstrate similar magnitudes of electron transfer rate constants due to comparable driving forces ($-\Delta G$) with co-initiators (**Fig. R5a**), it is indeed more accurate to consider the electron transfer process in terms of the actual rate ($v_{PET} (M\cdot s^{-1}) = k_{PET} [PC^*][Q]$), rather than the rate constant (k_{PET} , $M^{-1}\cdot s^{-1}$). From this standpoint, the concentrations of the PC in the excited state and the quencher are crucial in determining the electron transfer rate.

To this end, we performed a kinetic simulation based on the rate law to ascertain the concentration of PC in the excited

state under photostationary conditions, utilizing the reported Jablonski diagrams of the two PCs (*Adv. Mater.*, **35** 2204776 (2023); see **Fig. R5b**). The simulation results indicated that for 4Cz-IPN, the concentration ratio between the singlet and triplet excited states was approximately 100-fold, while for 4DP-IPN, this ratio exceeded 10,000-fold. This suggests that even with low concentrations of co-initiators, the triplet state of both PCs can contribute more significantly to the electron transfer process than the singlet state (*Nat. Commun.* **14**, 92 (2023); *Chem. Rev.* **122**, 1830–1874 (2022)). In our experimental setup, the concentrations of co-initiators, HNu 254 (5.2 mM) and Borate V (3.1 mM), are much lower than those used in photoluminescence quenching experiments. Consequently, the electron transfer from the triplet state is expected to be predominant in the actual polymerization process, in contrast to the singlet state, which would require significantly higher concentrations of co-initiators for quenching.

For clarification, we have added the following sentence in the revised manuscript as follows:

“4Cz-IPN, a cyanoarene-based thermally activated delayed fluorescent (TADF) material, generates singlet and triplet excited states with negligible energy differences,^{33,34} and its triplet state concentration is approximately 100 times higher, predominantly contributing to ET (Supplementary Fig. 4).”

and

“in the case of 4DP-IPN, the excited state triplet concentration is nearly 1000 times higher than that of the singlet, making the triplet the principal contributor, similar to 4Cz-IPN (see Supplementary Fig. 4).”

3) It's better to differentiate more for the terms “triplet-triplet energy transfer (EnT)” and “energy transfer (ET)”.

Response: Thanks for the reviewer's suggestion. However, we have chosen to retain the abbreviations 'EnT' and 'ET' for energy transfer and electron transfer, respectively, as they are commonly recognized in the field. To avoid confusion for the reader, we refrained from using abbreviations in the abstract and corrected any instances where they were used inaccurately or ambiguously within the text.

4) The authors achieved developing UV-blocking OCAs at a rate approximately 10 times faster than before. And claimed that the technique could be immediately commercialized. However, the manuscript's current presentation may leave readers seeking more compelling evidence to fully substantiate this claim.

Response: This information is proprietary to the company and cannot be fully disclosed. However, according to insights from Samsung Display and its partner companies, an integrated light dose of approximately 1500–3000 mJ cm⁻² is generally required to ensure the productivity of the film. This specification is corroborated by 3M product catalogs and various patents (**Fig. R6**). For instance, the recommended curing energy for 3M’s Liquid OCA 2321, as detailed in their guidelines, is 3000 mJ cm⁻² with irradiation at wavelengths ranging from 315 to 420 nm. Moreover, the feasibility of our approach is bolstered by a patent that has been filed for an acrylic resin curing system, which operates with a light dose of 2000–5000 mJ cm⁻² at 365 nm (*Patent JP 2020-46557 A*). Therefore, drawing from these precedents, we suggest that our OCA exhibits strong potential for successful commercialization. We have included the patent reference in the revised manuscript to underscore this potential.

3M™ Liquid Optically Clear Adhesive 2321		
Typical Cured Properties		
Note: The following technical information and data should be considered representative or typical only and should not be used for specification purposes.		
Chemistry	Acrylate	
Cure energy (mJ/cm ²)	3000	Radiometer, UVA
Refractive Index n_D	1.49	Abe, 25°C
Storage modulus (Pa), 25°C	1.30E+5	DMA, Shear Mode, 1 Hz
Color L*a*b* (corrected for substrate, 250um)	L* >99 a* <0.1 b* <0.5	Spectrophotometry, Lambda 950
Density (g/ml)	1.081	Pycnometry
Volume Shrinkage (%)	5.6% 4.7%	Pycnometry Water densitometer
Haze (%)	<1%	ASTM D1003
Transmittance (%) (average 380-780 nm)	>99%	ASTM E903
Adhesion to Glass (N/cm ²)	200	Pluck Testing

Fig. R6 Product catalog for 3M's OCA 2321, illustrating the cured properties of the adhesive. The catalog specifies the required light dosage for curing (3000 mJ cm⁻²), which is highlighted with a red line.

5) What about the UV blocking ability of the layers “BM and CF”, “ToE”, and “TFE”? In the dome of Fig. 4, it might be better to include those layers for the comparison.

Response: Thank you for the reviewer’s comments. We wish to clarify that the UV light passing through the color filter (CF) and reaching the pixel layer presents a challenge due to the color filter’s inadequate UV-blocking capability. Samsung Display has recognized this issue following the establishment of the color filter on encapsulation (CoE) technology and has tasked our lab with developing an OCA with UV-blocking abilities to address it.

To begin, the black matrix ("BM") containing carbon black can absorb UV light (as stated in *Patent JP 3508399B2*). However, since the BM's primary functions are to prevent color mixing and to enhance display resolution, it is positioned between RGB pixels (*Fig. R7a*). As a result, the BM layer does not efficiently protect RGB pixels from the vertical incidence of UV light. Secondly, the CF incorporates additives (such as pigments, dyes, solvents, and dispersants) and is typically produced using a UV-photolithography method (*ACS Photonics*, **6**, 3132 (2019) and *Appl. Opt.*, **59**, G137 (2020)) (*Fig. R7b*). Introducing UV-blocking capabilities into the CF layer may impede the UV-photolithography process, underscoring our manuscript's emphasis on the necessity of visible-light curing methods. The touch electrode (ToE) is processed atop the thin organic layer of the thin film encapsulation (TFE) using a patterning process for a metal mesh sensor (such as ITO, Cu, Ag, etc.) with a dielectric layer (*Patent US 20170153729A1*, *Patent US 20060097991A1*, and *International Journal of Precision Engineering and Manufacturing*, **16**, 2347 (2015)) (*Fig. R8a*). Although ITO in the ToE layer can absorb UV light in the range of 300 nm to 400 nm (*Fig. R8b*), its patterned mesh primarily covers electrode parts, not the RGB pixels. Thus, despite ITO's UV-blocking properties, it is likely inefficient in preventing RGB OLED degradation. Lastly, TFE layers serve to both flatten the pixel define layer (PDL) for a uniform surface and to protect RGB pixels from moisture and oxygen. Inorganic thin films in TFE layers, such as SiO_x, SiN_x, or SiO_xN_y, exhibit UV-blocking properties primarily for UV-B (290–320 nm) and UV-C (100–290 nm) regions (*Fig. R8c*). However, additional protection against UV-A (320–400 nm)—which penetrates the ozone layer, reaches the Earth's surface, and significantly impacts living organisms—is still necessary to enhance the lifespan of OLED devices.

While ideally, all layers would be integrated to create an OLED device for the UV-blocking test, we have prioritized testing the OCA layer in our laboratory due to accessibility and to simplify the experimental process. In order to commercialize the OCA we developed, we are currently planning to conduct a UV-blocking test on an actual device in cooperation with Samsung Display.

a ■ Device structure of a foldable display with UV-blocking optically clear adhesive (OCA) film

b ■ Photolithography-based MSFA fabrication process flow schematic

Fig. R7 (a) Schematic of a foldable display device structure featuring a UV-blocking optically clear adhesive (OCA) film. The acronyms TFT, BM, CF, ToE, TFE, and PDL stand for thin-film transistor, black matrix, color filter, touch-panel on encapsulation, thin-film encapsulation, and pixel define layer, respectively. (b) Process flow schematic for the UV-photolithography-based fabrication of multispectral filter arrays (MSFA). The contents of Fig. R7b have been reproduced from the cited publication (*ACS Photonics*, **6**, 3132 (2019)).

Fig. R8 (a) Schematic representation of touch-panel on encapsulation (ToE) and thin-film encapsulation (TFE) layers in an OLED device. (b-c) Optical properties of the components comprising the ToE and TFE layers: (b) UV/visible transmission spectra of materials (e.g., metal mesh and ITO) in the ToE layer, and (c) absorption spectra of SiO₂ in the TFE layer. The contents of Fig. R8(b-c) are reproduced from the cited publications (*International Journal of Precision Engineering and Manufacturing*, **16**, 2347 (2015) and *IOP Conference Series: Materials Science and Engineering*, **310**, 042029 (2019)).

6) For the adhesive test, please provide the testing conditions and the related standard. The peel strength in the newly proposed OCA is lower than in their previous work, what could be the potential reasons?

Response: Thank you for the reviewer's comments. In fact, we performed the peel strength test using an LS1 (AMETEK, USA) machine and adhered to the standard outlined in the Korean Industrial Standards (KS T 1028). A 10 kgf capacity load cell was employed, and the test specimens were secured using clamps designed for tensile testing. The acrylic syrup, cured between PETE film and release films, formed test specimens that were 1 cm in width. After the release films were removed, the adhesive film was bonded to the adherend using a 2 kg roller, passed over the film twice. Following a 24-hour bonding period, the peel strength of the tape-type adhesive was measured. This value was calculated as an average of the strength values recorded from 20% to 80% of the working range. We have updated the SI to include these detailed testing conditions as suggested by the reviewer as follow:

“The peel strength test was conducted with LS1 (AMETEK, USA) and followed the Korean Industrial Standards (KS T 1028). A load cell with a capacity of 10 kgf was used, and the test specimens were fixed by clamp used for a tensile test. For the peel test specimens, the acrylic syrup was cured between PETE film and release films, and the tested specimens were prepared as 1 cm wide. After removal of the release films from the prepared specimens, the OCA film was attached to the adherend using a 2 kg roller (2 round trips over the film). After 24 h of attachment time, the peel strength of tape-type adhesive was measured. Peel strength was obtained from the average of measured strength values from 20% to 80% of working range.”

The reviewer appears to have misconceived the adhesion values. Contrary to what was suggested, our data, as summarized in the accompanying **Table R2**, actually indicate that adhesion to glass was higher, while it was lower on CPI. The OCA developed in this study, intended for use with a cover window (ultrathin glass (UTG) in smartphones, and colorless polyimide (CPI) in larger displays such as tablets—as depicted in the device structure in **Fig. 1** of the manuscript, which has not yet been commercialized), may exhibit these variations in adhesion due to alterations in the monomer combination. In this study, we slightly adjusted the monomer mixture to further optimize the rheological properties, aiming to enhance the commercial viability of the OCA under development. Regardless, it's important to note that the adhesion levels typically required are in the range of approximately 1 to 4 N cm⁻¹. Therefore, the adhesion values we measured do not present a significant concern.

Table R2 Peel strength of the prepared OCAs film with different conditions

Entry	PC (ppm)	Monomers (molar ratio)	Co-initiators (ppm)	UVA (ppm)		Peel strength at 25 °C (N cm ⁻¹)	
				UVA-1	UVA-2	Glass	CPI
1 (Previous work)	4Cz-IPN (10)	[BA] : [HBA] = 4 : 1	DMAEAc (2000), DMAEA (3000)	1500 (0.3 phr)	4800 (1 phr)	2.4 (± 0.3)	5.4 (± 0.3)
2 (This work)	4DP-IPN (3)	[EHA] : [HBA] = 3 : 1	HNu 254 (1000), Borate V (600)	2000 (0.3 phr)	6400 (1 phr)	3.1 (± 0.2)	4.0 (± 0.1)

For the synthesis of acrylic syrup, all reaction mixtures were irradiated by 455 nm LEDs at RT and then the synthesized acrylic syrup was casted to a thickness of 50 μm between silicon-treated release films (100 μm-thick), then they were cured under 452 nm LEDs. The data of *Entry 1* have been reproduced from the cited publication (*Adv. Mater.*, **35**, 2204776 (2023)).

7) What advantages does the design of the OCA with UV-blocking ability offer over alternative strategies, like incorporating UVAs into the BM-CF layer? This alternative approach might potentially expedite OCA development significantly, given its potentially reduced demands on the choice of PC and initiators.

Response: As outlined in the manuscript, the technology we describe is known as *Color Filter on Encapsulation (CoE)* (<https://www.oled-info.com/samsung-display-announces-polarizer-free-eco2-oled-technology>). Developed to replace traditional polarizers, CoE aims to enhance energy efficiency. It has already been implemented in the Galaxy Z-Fold and is anticipated to be incorporated into future smartphones and other devices. Competitors of Samsung Display, *i.e.* BOE and Tianma, are also diligently working to refine this technology. Given that the CoE production process has been established and is currently scaled for mass production, the integration of a UV-blocking agent into the color filter may be technically feasible but could incur significant development and process costs. Consequently, this might lead to financial challenges. [REDACTED]

8) How does the UV-blocking ability depend on the thickness of the OCA?

Response: Thank you for the reviewer's constructive comments. The thickness of film layers such as cover film, OCA, and OLED in foldable displays typically ranges from 25 to 150 μm to ensure the necessary mechanical properties, including flexibility, as documented in *SID Symposium Digest of Technical Papers* **48**, 938–941 (2017) and the 3M Contrast Enhancement Film CEF36XX Series. Accordingly, we prepared acrylic OCAs with a thickness of approximately 50 μm in our laboratory. We acknowledge the challenge of consistently producing OCAs thinner than 50 μm . In response to the reviewer's comment, we evaluated the UV-blocking abilities of OCAs at various thicknesses by measuring UV/Vis transmittance using absorption spectroscopy (**Fig. R9**). We found that all films, irrespective of thickness, effectively block UV light below 400 nm. However, thicker films demonstrated lower transmittance in the visible light spectrum above 400 nm. Consequently, taking into account the requirements for display application efficiency, we have identified 50 μm as the optimal thickness for our OCAs. These insights into the correlation between the UV-blocking ability of OCAs and their thickness have been added to the revised manuscript and SI as follows:

*“Given that the UV-blocking ability did not significantly differ across the OCA thickness range of 50–200 μm , we proceeded with a 50 μm thickness for subsequent experiments, aligning with the standard for foldable OCA films (see **Supplementary Fig. 14a**).”*

Fig. R9 UV/Vis Transmittance of UV-Blocking OCAs at various thicknesses. To prepare the OCAs ([EHA]:[HBA] = 3:1), we incorporated 4DP-IPN (3 ppm), HNu 254 (1000 ppm), Borate V (600 ppm), UVA-1 (2000 ppm), and UVA-2 (6400 ppm). The UV/Vis transmittance of the OCA films was measured across thicknesses ranging from 50 to 200 μm .

9) The misspelling of 'TFE' is found in the Fig.1 legend.

Response: We have properly addressed in the revised manuscript.

REVIEWER COMMENTS

Reviewer #1 (Remarks to the Author):

The authors responded to my question very diligently, and I am okay with their answers.

Reviewer #2 (Remarks to the Author):

Most of my comments and suggestions are not addressed properly. I can't recommend accepting this manuscript at the current version.

1. For my previous Comment 1.

There is still major misleading information in the manuscript, including the abstract, introduction, Figs., etc., regarding distinguishing the innovation in the current work from their previous work. Again, I would like to point out that the photocatalyst (PC), 4DP-IPN, is not new. However, the authors still emphasize that "We have re-engineered the photocatalyst (PC)..." and "we delicately modified the structure of the existing PC to design a new PC, 4DP-IPN,...". In addition, Figures 1 and 2 still contain a large amount of redundant information from their earlier work.

2. For my previous Comment 2.

The formation of excitons during photoexcitation primarily consists of singlet excitons, a well-established fact within the photophysical community. However, the authors' calculation results appear to draw the opposite conclusion, suggesting that triplet excitons, rather than singlet excitons, dominate the photocatalyst. (Note: There seems to be a discrepancy or confusion regarding whether singlet or triplet excitons are dominant in the author's Response)

The author adopted the calculation method used in their previous work (*Adv. Mater.*, 35 2204776 (2023)), which is likely flawed.

a. The authors calculated the concentrations of [S1] and [T1] based on the equations S1 and S2 in the Supplementary Information. The rate constants are shown in Table S3. However, the k_r , $T1$ and k_{nr} , $T1$ are not available in the Table, how did the authors do the calculations?

b. The calculated results might be meaningless to estimate the concentrations of [S1] and [T1] in the photo-initiating system (PIS) if the equations neglect the electron and energy transfer between the PCs and the acceptors (photoinitiator and UVAs). The authors mentioned that the rate constant of

photoinduced ET (PET) between 4Cz-IPN and DMAEAc is $k_{PET} = 2.1 \times 10^7 \text{ M}^{-1} \text{ s}^{-1}$, which means that, at least in the time range longer than $\sim 5\mu\text{s}$, the concentrations of [S1] and [T1] are largely influenced by the PET. Not to mention the singlet energy transfer rate, which might be faster. If the electron and energy transfers between the PCs and the acceptors (photoinitiator and UVAs) are ignored, only the results of $t=0$ are useful. BTW, why is the ratio between the concentration of [S1] and [T1] at $t=0$ not equal to 1 for some PCs, e.g., 4DP-IPN, and 4-p,p-DCDP-IPN?

3. For my previous Comment 3: It's better to differentiate more for the terms "triplet-triplet energy transfer (EnT)" and "energy transfer (ET)".

The authors retained the abbreviations 'EnT' and 'ET' for energy transfer and electron transfer, respectively. Why are the abbreviations the authors retained not the same with their previous version of manuscript? And how did they do the improvement, and where?

4. For my previous Comment 4

The current work employs visible light to cure the optically clear adhesive (OCA) and claims that the technique could be immediately commercialized. I asked why, and the authors responded their approach is bolstered by a patent that has been filed for an acrylic resin curing system, which operates with a UV light source, which means the technique can't be commercialized with visible light? I'm still not convinced how this technique could be immediately commercialized in the current presentation of the manuscript. In the author's response to my previous Comment 5, the authors mentioned that "Introducing UV-blocking capabilities into the CF layer may impede the UV-photolithography process,...". But introducing UV-blocking capabilities into the OCA is OK for the UV curing process? If so, it is still feasible to introduce UV-blocking capabilities into the CF layer regarding my previous Comment 5 and Comment 7?

5. For my previous Comment 5

It would be highly valuable to also include the UV blocking ability of the CF layer or at least have a discussion in the manuscript.

Lastly, but certainly not least, it is imperative to conduct the discussions based on the facts rather than relying on statements or opinions from Samsung Display.

Reviewer #2 (Remarks to the Author):

Most of my comments and suggestions are not addressed properly. I can't recommend accepting this manuscript at the current version.

Response: We regret that our initial revision did not fully meet the expectations of the reviewer. In this current revision, we have made every effort to address the reviewer's feedback comprehensively. Specifically, we have eliminated phrases and sentences that might lead to an overestimation of our results, thereby more clearly highlighting the study's importance and novelty. We have also addressed the reviewers' technical questions and requests in detail. We are deeply appreciative of the reviewers' time and effort dedicated to our manuscript and believe that these contributions have significantly enhanced its quality.

1. For my previous Comment 1, In the Abstract and Introduction, the authors assert that they employed a newly designed PC (Photocatalyst), specifically 4DP-IPN. However, it is worth noting that this PC, 4DP-IPN, has been previously explored in their earlier research. Throughout the manuscript, substantial effort is invested in comparing 4DP-IPN with 4Cz-IPN, both in the main text and Figure 1. This extensive comparison may inadvertently obscure the innovative aspects and significance of the current study. It is essential to emphasize that the primary distinction between the photo-initiating systems (PIS) developed in this work and their prior research lies in the choice of the photoinitiator. This pivotal distinction could be highlighted more prominently in the present manuscript. Additionally, Figures 1 and 2 contain a large amount of redundant information from their earlier work, warranting attention for a more concise and focused presentation.

There is still major misleading information in the manuscript, including the abstract, introduction, Figs., etc., regarding distinguishing the innovation in the current work from their previous work. Again, I would like to point out that the photocatalyst (PC), 4DP-IPN, is not new. However, the authors still emphasize that “We have re-engineered the photocatalyst (PC)...” and “we delicately modified the structure of the existing PC to design a new PC, 4DP-IPN,...”. In addition, Figures 1 and 2 still contain a large amount of redundant information from their earlier work.

Response: Responding to the reviewer's comments, we have revised or removed certain sentences and phrases in the introduction and abstract that could lead to an overvaluation of our results. As the reviewer correctly noted, the catalyst 4DP-IPN has been developed and utilized in previous studies. Therefore, to avoid reader confusion, we have replaced the term 'reengineered' with 'selected in our previously constructed catalyst library.' Additionally, to more clearly convey the novelty of this study, we slightly modified **Figure 1b** and relocated **Figure 1c** from the main figure to the supplementary information. Regarding **Figure 2a**, which depicts a mechanism already proposed in the previous paper, we considered its removal. However, we chose to keep it to aid comprehension for readers who may not be familiar with our prior work, thereby eliminating the need for them to refer to earlier publications.

#Previous version: “We have re-engineered the photocatalyst (PC) to minimize electron and energy transfer to UV-blocking agents and introduced new co-initiators ~”

#Revised version: “We have carefully selected the photocatalyst (PC) to minimize electron and energy transfer to UV-blocking agents and have chosen co-initiators ~”

#Previous version: “Here, we develop a highly efficient PIS that operates under visible-light irradiation, allowing to

produce UV-blocking OCAs at a fast enough rate for commercialization. Through a mechanistic analysis of existing PIS, we found that electron transfer (ET) and energy transfer (EnT) between the PC and the UVAs ~”

#Revised version: “Here, we have developed a highly efficient PIS that operates under visible-light irradiation, allowing for the production of UV-blocking OCAs at a much faster rate compared to the previously reported PIS.³¹ Through a mechanistic analysis of existing PIS, we found that electron transfer (ET) and triplet-triplet energy transfer (EnT) between the PC and the UVAs ~”

#Previous version: “In pursuit of resolving the issue, we employed the systematic molecular catalyst design principles proposed in our previous study.³⁴ This methodology utilizes ~”

#Revised version: “To resolve the issue, we screened the PC candidates established in our previous study, utilizing the systematic molecular catalyst design platform.³⁴ This platform enables ~”

#Comparison of original and revised figures:

2. For my previous Comment 2, The molecules are initially excited to the singlet state and subsequently transition into the triplet state in the photoexcitation. The emphasis in this work primarily centers on the design principles pertaining to the triplet state of various species within the photo-initiating system (PIS). However, the question arises: why is the singlet state of these materials not equally critical and deserving of discussion?

The formation of excitons during photoexcitation primarily consists of singlet excitons, a well-established fact within the photophysical community. However, the authors' calculation results appear to draw the opposite conclusion, suggesting that triplet excitons, rather than singlet excitons, dominate the photocatalyst. (Note: There seems to be a discrepancy or confusion regarding whether singlet or triplet excitons are dominant in the author's Response). The author adopted the calculation method used in their previous work (Adv. Mater., 35 2204776 (2023)), which is likely flawed.

Response: We are grateful for the reviewer's comments, particularly regarding a central point that could potentially lead to misunderstandings and thus requires further clarification. As the reviewer notes, organic dyes are typically excited to the singlet excited state manifold (S_n), which then rapidly deactivates to S_1 . This is indeed the primary step occurring in our dyes. However, our photocatalysts (PCs) differ from classic dyes as they are based on thermally activated delayed fluorescence (TADF). In this process, T_1 is efficiently populated from S_1 via intersystem crossing (ISC), and subsequently, S_1 is repopulated from T_1 via reverse ISC (RISC), resulting in a cycling effect. This mechanism leads to a prolonged excited state lifetime, making these compounds particularly effective as photocatalysts (see *Chem. Soc. Rev.*, 50, 7587 (2021)). Our primary strategy focuses on the utilization of the triplet excited state in photocatalysis for preparing OCA films. The longer lifetime of the triplet state enables a higher effective PC^* concentration, thereby reducing the required PC loading compared to organic dyes that predominantly generate the S_1 state. To illustrate this, we conducted a kinetic simulation for the exciton population, enabling us to quantitatively evaluate the contributions of each excited state in the electron transfer process. Detailed explanations of the kinetic simulations are provided below.

a. The authors calculated the concentrations of $[S_1]$ and $[T_1]$ based on the equations S1 and S2 in the Supplementary Information. The rate constants are shown in Table S3. However, the k_r, T_1 and k_{nr, T_1} are not available in the Table, how did the authors do the calculations?

Response: The rate constants k_{r, T_1} and k_{nr, T_1} were in fact neglected in the calculations; this approximation is justified for so-called "strong TADF emitters", as shown earlier by others and by us (*J. Phys. Chem. A* 117, 5607–5612 (2013), *Chemical Society Reviews*, 50, 7587-7680 (2021), *The Chemical Record*, 20, 831-856 (2020) and *Nature Commun.* 14, 92 (2023)). The condition for strong TADF emitters is fulfilled for compounds with (i) a high total fluorescence quantum yield Φ_F and, at the same time, (ii) a high fraction of delayed fluorescence Φ_{DF}/Φ_F . Both conditions are in fact fulfilled for 4DP-IPN (*Macromolecules*. 52, 5538 (2019)). In this case, the triplet deactivation is largely dominated by RISC. For a comprehensive understanding, we now added a detailed description of the process to obtain each rate constant in the Jablonski diagram of the PC in the revised Supplementary materials with the corresponding references:

■ Evaluation of k_{r, S_1} , k_{nr, S_1} , k_{ISC} and k_{RISC} .

The general TADF kinetics can be obtained from the differential equations for the singlet and triplet excited state (S_1 and T_1) deactivation.

$$\frac{d[S_1]}{dt} = -k_s \cdot [S_1] + k_{RISC} \cdot [T_1] + \alpha \cdot I \quad (5)$$

$$\frac{d[T_1]}{dt} = k_{ISC} \cdot [S_1] - k_T \cdot [T_1] \quad (6)$$

where $k_s = k_{r, S_1} + k_{nr, S_1} + k_{ISC}$, $k_T = k_{r, T_1} + k_{nr, T_1} + k_{RISC}$, I is the intensity of excitation, and α is the absorption coefficient; the solutions are described below

$$S_1(t) = \frac{S_1(0)}{A_2 - A_1} [(A_2 - k_s) \exp(-A_1 t) + (k_s - A_1) \exp(-A_2 t)] \quad (7)$$

$$T_1(t) = \frac{S_1(0) \cdot k_{ISC}}{A_2 - A_1} [\exp(-A_1 t) - \exp(-A_2 t)] \quad (8)$$

and the total intensity is obtained as

$$I(t) = k_{r,S_1}S_1(t) + k_{r,T_1}T_1(t) \quad (9)$$

$$I(t) = \frac{S_1(0)}{A_2 - A_1} [(k_{r,S_1}(A_2 - A_1) + k_{r,T_1}k_{ISC}) \exp(-A_1 t) + (k_{r,S_1}(k_S - A_1) - k_{r,T_1}k_{ISC}) \exp(-A_2 t)] \quad (10)$$

The exponents $A_{1,2}$ (which correspond to the reciprocal values of the prompt/delayed PL lifetime constants, i.e., τ_{PF}^{-1} and τ_{DF}^{-1} , respectively) are given by

$$A_{1,2} = \frac{1}{2} (k_S + k_T \mp (k_T - k_S) \sqrt{1 + 4 \cdot k_{ISC}k_{RISC}/(k_T - k_S)^2}) \quad (11)$$

The total PL quantum yield Φ_{PL} is given as the sum of fluorescence and phosphorescence quantum yields ($\Phi_{PL} = \Phi_F + \Phi_{PH}$), where Φ_F consists of a prompt fluorescence (Φ_{PF}) and delayed fluorescence (Φ_{DF}) part; the prompt part is defined by

$$\Phi_{PF} = \frac{k_{r,S_1}}{k_S} \quad (12)$$

In the presence of large number of TADF cycles, the total Φ_F under steady state is obtained as,¹⁴

$$\Phi_F = \Phi_{PF} \frac{1}{(1 - \eta_{ISC} \cdot \eta_{RISC})} \quad (13)$$

where $\eta_{ISC} = k_{ISC}/k_S$ and $\eta_{RISC} = k_{RISC}/k_T$ are the efficiencies for ISC and RISC, respectively. Similarly, Φ_{PH} is obtained as

$$\Phi_{PH} = \frac{\eta_{ISC}k_{PH}}{k_T(1 - \eta_{ISC} \cdot \eta_{RISC})} \quad (14)$$

The condition for strong TADF emitters (i.e. large Φ_F and large Φ_{DF}/Φ_F) emitters translates to $k_{RISC} \gg k_{r,T_1}, k_{nr,T_1}$, so that $\eta_{RISC} \approx 1$; this simplifies equation (13) to

$$\Phi_F = \frac{\Phi_{PF}}{(1 - \eta_{ISC} \cdot \eta_{RISC})} \approx \frac{\Phi_{PF}}{(1 - \eta_{ISC})} = \frac{k_S \Phi_{PF}}{(k_{r,S_1} + k_{nr,S_1})} = \frac{k_{r,S_1}}{(k_{r,S_1} + k_{nr,S_1})} \quad (15)$$

Furthermore, for TADF compounds with a non-negligible ΔE_{ST} , RISC is much smaller than ISC (i.e., $k_{RISC} \ll k_{ISC}$).² Under these conditions, with a Taylor expansion ($y = \sqrt{1+x} \approx 1 + \frac{x}{2}$ for $x \ll 1$), the solutions for $A_{1,2}$ simplify to

$$A_1 = \tau_{PF}^{-1} = \frac{1}{2} (k_S + k_T - (k_T - k_S) \sqrt{1 + 4 \cdot k_{ISC}k_{RISC}/(k_T - k_S)^2}) = k_S - \frac{k_{ISC}k_{RISC}}{k_T - k_S} \approx k_S \quad (16)$$

$$A_2 = \tau_{DF}^{-1} = \frac{1}{2} (k_S + k_T + (k_T - k_S) \sqrt{1 + 4 \cdot k_{ISC}k_{RISC}/(k_T - k_S)^2}) = k_T \left(1 - \frac{k_{ISC}k_{RISC}}{k_T k_S}\right) \approx k_{RISC}(1 - \eta_{ISC}) \quad (17)$$

Finally, the radiative rate constant k_{r,S_1} can be estimated from the Strickler-Berg formula, which in its simplified form reads,^{15,16}

$$k_{r,S_1,SB} = 0.667(s^{-1}cm^2) \frac{E_{F,vert}^3}{E_{A,vert}} n^2 f = 4.34 \cdot 10^7 (s^{-1}eV^{-2}) \frac{E_{F,vert}^3}{E_{A,vert}} n^2 f \quad (18)$$

where f is the TD-DFT calculated oscillator strength of vertical absorption, n is the refractive index of solvent and E is the energy of vertical absorption and emission respectively for the lowest energetic CT transition.

In summary, the photophysical rate constants of PCs in Jablonski diagram were evaluated by experimental (i.e., prompt/delayed fluorescence decays) and computational method (i.e., TD-DFT), which each relation is simplified to

$$k_{r,S_1} = 4.34 \cdot 10^7 (s^{-1} eV^{-2}) \frac{E_{F,vert}^3}{E_{A,vert}} n^2 f \quad (19)$$

$$k_{nr,S_1} = \frac{k_{r,S_1}}{\Phi_F} - k_{r,S_1} = k_{r,S_1} \left(\frac{1}{\Phi_F} - 1 \right) \quad (20)$$

$$k_{ISC} = \tau_{PF}^{-1} - \frac{k_{r,S_1}}{\Phi_F} \quad (21)$$

$$k_{RISC} = \frac{\tau_{DF}^{-1}}{1 - k_{ISC} \tau_{PF}} \quad (22)$$

where f , n , E , Φ_F , τ_{PF} , and τ_{DF} have been defined earlier.

b. The calculated results might be meaningless to estimate the concentrations of [S1] and [T1] in the photo-initiating system (PIS) if the equations neglect the electron and energy transfer between the PCs and the acceptors (photoinitiator and UVAs). The authors mentioned that the rate constant of photoinduced ET (PET) between 4Cz-IPN and DMAEAc is $k_{PET} = 2.1 \times 10^7 \text{ M}^{-1} \text{ s}^{-1}$, which means that, at least in the time range longer than $\sim 5 \mu\text{s}$, the concentrations of [S1] and [T1] are largely influenced by the PET. Not to mention the singlet energy transfer rate, which might be faster. If the electron and energy transfers between the PCs and the acceptors (photoinitiator and UVAs) are ignored, only the results of $t=0$ are useful. BTW, why is the ratio between the concentration of [S1] and [T1] at $t=0$ not equal to 1 for some PCs, e.g., 4DP-IPN, and 4-p,p-DCDP-IPN?

Response: We appreciate the reviewer for the constructive comments. The primary aim of our kinetics simulations is to estimate the kinetics of radical production in each photoinitiation system (PIS), as clearly depicted in **Figure 3e** in the original manuscript. These calculations take into account all factors that could potentially affect radical generation for photopolymerization. This includes the concentration of the excited state of the photocatalyst (PC), the rates of electron and energy transfer between the PC and the co-initiator, and between the PC and UV absorbers (UVAs). By considering these factors, we were able to determine the efficiency with which each PIS can generate radicals, thereby influencing the speed of curing.

However, this simulation does not intuitively explain why PISs exhibit such radical generation kinetics. We aimed to clarify this through **Figure 3d**. Specifically, the cyanoarene-based PCs used in our study possess TADF properties, which efficiently generate triplet excited states. The extent to which these triplet excited states are formed significantly influences the overall curing speed. The energy transfer from the PCs' triplet states to the UVAs competes with the electron transfer to the co-initiators, impacting the curing speed. However, as the reviewer pointed out, the energy or electron transfer from PCs to the UVAs or the co-initiator is indeed competitive. Calculating the concentration of the triplet excited state without considering the co-initiators might lead to misunderstandings or confusion among readers. To address this, we provided data on the amount of triplet excited state generated solely from the PCs, in the absence of UVA and co-initiator (see below). This approach allows readers to more clearly comprehend the differences between 4DP-IPN and 4Cz-IPN.

#Comparison of original and revised figures:

#Issues about singlet energy transfer: We can rule out the singlet-singlet energy transfer from PCs to UVAs, as this process is substantially endothermic given their respective singlet energies. Specifically, the singlet energies are E_{00} (S_1 :UVA-1) = 3.08 eV and E_{00} (S_1 :UVA-2) = 4.02 eV for UVAs, and E_{00} (S_1 :4Cz-IPN) = 2.77 eV and E_{00} (S_1 :4DP-IPN) = 2.55 eV for PCs, as shown in *Fig. R1*.

Fig. R1. Chemical structures and photophysical properties of UVAs including UV/vis absorption and photoluminescence (PL) emission spectra in ethyl acetate ($1.0 \times 10^{-5} \text{ M}$). $E(S_1)$ and $E(T_1)$ were extracted from the onset of PL and onset of gated PL in ethyl acetate at 65 K, respectively. The onsets were obtained by tangential method, i.e., the intersection of the tangent, set at the high energy slope of the spectrum, with the x-axis. Because the phosphorescence of UVA-2 was not observed, therefore $E(T_1)$ of UVA-2 was referred to the literature, where $E(T_1)$ was estimated from the phosphorescence quenching experiments with the quencher having $E(T_1) = 2.59 \text{ eV}$ (*Journal of Luminescence* 166, 203–208 (2015)). $E(T_1)$ of UVAs calculated by TD-DFT were also given.

#Issues about $[S_1]$ and $[T_1]$ ratio: The ratio between of $[T_1]$ and $[S_1]$ of strong TADF-emitting PCs (e.g., 4DP-IPN, and 4-p,p-DCDP-IPN) is immediately increased nanoseconds after light absorption of the PC with very fast intersystem crossing ($k_{\text{ISC}} \sim 10^8 \text{ s}^{-1}$) from S_1 to T_1 compared to reverse intersystem crossing ($k_{\text{RISC}} \sim 10^4 \text{ s}^{-1}$) from T_1 to S_1 .

3. For my previous Comment 3: It's better to differentiate more for the terms "triplet-triplet energy transfer (EnT)" and "energy transfer (ET)".

The authors retained the abbreviations 'EnT' and 'ET' for energy transfer and electron transfer, respectively. Why are the abbreviations the authors retained not the same with their previous version of manuscript? And how did they do the improvement, and where?

Response: We regret that we misunderstood the reviewer's question, leading to confusion. As the reviewer correctly noted, we have consistently used 'ET' to denote electron transfers and 'EnT' for energy transfers, a practice we've maintained since our early works. We acknowledge that the term 'energy transfer (ET)' in the original abstract was a typo, and we corrected this error in the previous revision. To avoid further misunderstandings, in this revision, we have replaced the term 'energy transfer' with 'triplet-triplet energy transfer' throughout the document.

4. For my previous **Comment 4**, the current work employs visible light to cure the optically clear adhesive (OCA) and claims that the technique could be immediately commercialized. I asked why, and the authors responded their approach is bolstered by a patent that has been filed for an acrylic resin curing system, which operates with a UV light source, which means the technique can't be commercialized with visible light? I'm still not convinced how this technique could be immediately commercialized in the current presentation of the manuscript. In the author's response to my previous **Comment 5**, the authors mentioned that "Introducing UV-blocking capabilities into the CF layer may impede the UV-photolithography process,...". But introducing UV-blocking capabilities into the OCA is OK for the UV curing process? If so, it is still feasible to introduce UV-blocking capabilities into the CF layer regarding my previous **Comment 5** and **Comment 7**?

Reviewer's previous Comment 4) The authors achieved developing UV-blocking OCAs at a rate approximately 10 times faster than before. And claimed that the technique could be immediately commercialized. However, the manuscript's current presentation may leave readers seeking more compelling evidence to fully substantiate this claim.

Reviewer's previous Comment 5) What about the UV blocking ability of the layers "BM and CF", "ToE", and "TFE"? In the dome of Fig. 4, it might be better to include those layers for the comparison.

Reviewer's previous Comment 7) What advantages does the design of the OCA with UV-blocking ability offer over alternative strategies, like incorporating UVAs into the BM-CF layer? This alternative approach might potentially expedite OCA development significantly, given its potentially reduced demands on the choice of PC and initiators.

Response: Thank you for the reviewer's insightful questions. These inquiries will significantly aid readers in understanding our paper without confusion. The reviewer posed three distinct questions, which we will address individually below.

(Response to comment #4) The film produced in this study meets the commercialization criteria set by Samsung Display. Therefore, it can be commercialized immediately once the challenges of mass production are resolved. As you may know, each company has different standards for film properties for commercialization, and these are often trade secrets, so we are unable to disclose them precisely. Direct investment in mass production of OCA film is risky and time-consuming, especially in setting up production facilities. [REDACTED]

Thus, whether our product will be used in actual display devices depends on many factors beyond just the technology itself. We cannot disclose all these points in the paper. Therefore, we have removed the phrase 'readily commercially available' and moderated the language accordingly.

Previous version: “Here, we develop a highly efficient PIS that operates under visible-light irradiation, allowing to produce UV-blocking OCAs at a fast enough rate for commercialization.”

Revised version: “Here, we have developed a highly efficient PIS that operates under visible-light irradiation, allowing for the production of UV-blocking OCAs at a much faster rate compared to the previously reported PIS.”

Previous version: “Moreover, we successfully demonstrated their practical UV-blocking capabilities by applying them to OLED devices, indicating their potential for immediate commercialization.”

Revised version: “Moreover, we successfully demonstrated their practical UV-blocking capabilities by applying them to OLED devices, indicating their potential for various applications that require UV-blocking abilities.”

(Response to comment #5 and #7) The confusion may have arisen due to insufficient information provided about the display device fabrication process. As illustrated below, the production of a display device involves separately manufacturing the display panel and the cover window, and then combining them. The UV-blocking OCA developed in this study represents one layer of the cover window, while the color filter constitutes one layer of the panel (**Fig. R2**). Therefore, the production of UV-blocking OCA is feasible, independent of the color filter issue. To prevent further confusion, we have modified **Figure 1a** as follows.

Fig. R2. Device structure of a foldable display with UV-blocking optically clear adhesive (OCA) film; TFT, BM, CF, ToE, TFE and PDL denote thin film transistor, black matrix, color filter, touch-panel on encapsulation, thin film encapsulation and pixel define layer, respectively. Schematic illustration for process of layer assembly is also given.

Additionally, for your reference, we have provided a link to a simplified overview of the general process for creating a color filter (<https://www.toppan.com/en/electronics/colorfilter/display/production/>). As the reviewers are aware, color filters are designed for color purity and incorporate dyes that absorb visible light. Consequently, visible light curing is not feasible, and UV curing is typically employed. Therefore, adding a UV blocking agent to this process is challenging and would likely require extensive optimization. Consequently, we believe that developing UV blocking OCA represents a more cost-effective approach.

#Comparison of original and revised figures:

5. For my previous **Comment 5**, it would be highly valuable to also include the UV blocking ability of the CF layer or at least have a discussion in the manuscript. Lastly, but certainly not least, it is imperative to conduct the discussions based on the facts rather than relying on statements or opinions from Samsung Display.

Reviewer's previous Comment 5) What about the UV blocking ability of the layers "BM and CF", "ToE", and "TFE"? In the dome of Fig. 4, it might be better to include those layers for the comparison.

Response: Following the reviewer's argument, we added the following sentences to the introduction and results parts: *"Additionally, introducing UV-blocking properties to the color filter might appear viable, but it is impractical for two main reasons: i) these filters are usually produced using a UV curing process,^{9,10} incompatible with the addition of UV absorbers (UVAs), and ii) visible-light curing, while a potential alternative, is ineffective because color filters inherently absorb visible light."* And *"Ideally, for UV blocking testing, all layers, including the color filters, should be integrated to create an OLED device. However, due to practical difficulties in fabricating color filters in the laboratory, we opted for a simplified device structure. This approach is justified, given that color filters block very little UV light."*

REVIEWER COMMENTS

Reviewer #2 (Remarks to the Author):

See attached.

The further revisions addressed most of the questions, making it more ready for publication. But I still have some remaining questions and comments:

1. For the calculation of the concentrations of [S1] and [T1]:

a) $k_{r,T1}$ and $k_{nr,T1}$ were neglected in the calculations. I agree with the authors that those two neglects are reasonable for strong TADF emitters with two requirements: a high Φ_F and a high Φ_{DF}/Φ_F . However, the Φ_F of 4DP-IPN is only about 26% (Adv. Mater., 35 2204776 (2023)), which is not high at all. Besides, the Φ_{RISC} of 4DP-IPN is as low as $\sim 10^4 \text{ s}^{-1}$, which is notably lower than the strong TADF emitters like 4CZ-IPN, indicating the triplet excitons are more likely to be quenched through processes like TTA.

There was a different model (Nat. Mater. 14, 330, (2015)), which neglected the $k_{nr,S1}$, for estimating the kinetic processes in TADF emitters. It was suggested that for non-efficient TADF emitters, e.g., 4DP-IPN, the $k_{nr,T1}$ is much faster than the k_{RISC} , implying that non-radiative decay from T1 dominates over the RISC process back to the singlet state. Therefore, it is not appropriate to neglect the contribution of $k_{nr,T1}$ to estimate the k_T in equation 6.

Since the dominating energy transfer process in the photo-initiating systems is the triplet-triplet energy transfer, it is crucial to include the $k_{nr,T1}$ in equation 6. Therefore, it would be more rigorous to do the calculations using the models that consider the $k_{nr,T1}$ (Nat. Mater. 14, 330, (2015)) or both the $k_{nr,S1}$ and $k_{nr,T1}$ (Nat. Photonics. 6, 253, (2012)).

In the meantime, since the k_{PET} between PCs and UVAs is estimated by the authors to be around $2.1 \times 10^7 \text{ M}^{-1} \text{ s}^{-1}$, which is around 3 orders higher than either $k_{r,T1}$ or $k_{nr,T1}$. So, in equation 6, calculating T1 concentration without considering k_{PET} is meaningless to explain why 4DP-IPN is a better PC than 4CZ-IPN (as shown in Fig. R1).

b) I noticed an extra term αI in equation 5 compared to the authors' previous calculations (Adv. Mater., 35 2204776 (2023))? It seems equations 7 and 8 are derived as solutions to equations 5 and 6 without considering the αI term? BTW, what is the A_s in equation 10?

2. As the authors suggested, the OCA is compatible with the UV curing process, which implies that adding UVAs in color filters would not prevent their UV curing process. However, it would be helpful for readers to better appreciate the rationale behind adding UVAs to the OCA if the authors could discuss the UV-blocking ability of the color filters.

Reviewer #2 (Remarks to the Author):

The further revisions addressed most of the questions, making it more ready for publication. But I still have some remaining questions and comments:

Response: We sincerely appreciate the very positive response from the reviewers. In this revised version, we have addressed reviewer's remaining questions and comments, aiming to provide readers with a more comprehensive understanding of our work. We highlighted our revision in main manuscript and supplementary materials as blue color. We are grateful for the continued efforts that the reviewers have dedicated to our work to significantly improve its quality.

1. For the calculation of the concentrations of [S1] and [T1]:

a) $k_{r,T1}$ and $k_{nr,T1}$ were neglected in the calculations. I agree with the authors that those two neglects are reasonable for strong TADF emitters with two requirements: a high Φ_F and a high τ_{DF}/τ_F . However, the Φ_F of 4DP-IPN is only about 26% (Adv. Mater., 35, 2204776 (2023)), which is not high at all. Besides, the Φ_{RISC} of 4DP-IPN is as low as $\sim 10^4$ s⁻¹, which is notably lower than the strong TADF emitters like 4CZ-IPN, indicating the triplet excitons are more likely to be quenched through processes like TTA.

There was a different model (Nat. Mater. 14, 330, (2015)), which neglected the $k_{nr,S1}$, for estimating the kinetic processes in TADF emitters. It was suggested that for non-efficient TADF emitters, e.g., 4DP-IPN, the $k_{nr,T1}$ is much faster than the k_{RISC} , implying that non-radiative decay from T₁ dominates over the RISC process back to the singlet state. Therefore, it is not appropriate to neglect the contribution of $k_{nr,T1}$ to estimate the k_T in equation 6.

Since the dominating energy transfer process in the photo-initiating systems is the triplet-triplet energy transfer, it is crucial to include the $k_{nr,T1}$ in equation 6. Therefore, it would be more rigorous to do the calculations using the models that consider the $k_{nr,T1}$ (Nat. Mater. 14, 330, (2015)) or both the $k_{nr,S1}$ and $k_{nr,T1}$ (Nat. Photonics. 6, 253, (2012)).

Response: We thank the reviewer for the thoughtful comment. In response to the reviewer's comment for triplet quenching process, we used extremely low PC concentrations in the range of $1.5\text{--}5.1 \times 10^{-5}$ M (for 3–10 ppm PC loading) in the polymerization, thus bimolecular collisions between PC* species to occur triplet-triplet annihilation (TTA) are negligibly slow. Nevertheless, we normally take into account the delayed fluorescence lifetime to determine the triplet decay process of PCs itself. Therefore, it is plausible to regard undesired quenching process (e.g. TTA) as inherently encompassed within delayed fluorescence lifetime.

The overall Φ_F of 4DP-IPN in ethyl acetate (EA; which was used in the current study), $\Phi_F = 26\%$ under purged conditions, is much lower than for instance in CH₃CN or DMSO, reported by us earlier ($\Phi_F = 62\%$ in CH₃CN, Nat. Commun. 14, 92 (2023)). In any case, the neglect of $k_{nr,T1}$ is not based on the question of an overall high Φ_F , but on dominance of delayed fluorescence (i.e. Φ_{DF}). Thus, the ratio of delayed fluorescence vs. prompt fluorescence should be large; in fact, it was shown earlier that the approximation should hold for $\Phi_{DF}/\Phi_{PF} > 4$ (Methods Appl. Fluoresc. 5, 012001 (2017)). As shown in Fig. R1, in the current case, we have $\Phi_{DF}/\Phi_{PF} = 0.22/0.04 = 5.5$; thus, the approximation should be valid. Further support is given by Fig. R2, comparing the ratio of exciton populations between two assumptions $k_{nr,T1} \sim 0$ and $k_{nr,S1} \sim 0$, which did not qualitatively change the scenario of exciton population.

Fig. R1. (a) PLQY of 4DP-IPN and 4Cz-IPN in ethyl acetate. PLQY values (Φ_F) of both PCs were relatively measured against coumarin C153, and PLQY values of prompt fluorescence and delayed fluorescence were determined from the total PLQY and the proportion of the integrated area of individual components in the TCSPC spectra to the total integrated area (*Chemical Engineering Journal* 416, 129097 (2021)). (b) Ratio of the rate constants of RISC (k_{RISC}) between assuming $k_{nr,S1} = 0$ and $k_{nr,T1} = 0$ for each PLQY with color properties indicating the ratio of Φ_{PF} and Φ_{DF} ; solid black line, $\Phi_{PF}/\Phi_{DF} = 1$. All contents of **Fig. R1b** are reproduced from cited publications (*J. Phys. Chem. A* 125, 8074–8089 (2021)).

Fig. R2. Comparison of kinetic simulation for exciton population between two assumptions; (a) under assumption of $k_{nr,T1} \sim 0$ and $k_{r,T1} \sim 0$ with photophysical values obtained in this work and (b) under assumption of $k_{nr,S1} \sim 0$ and $k_{r,T1} \sim 0$ with photophysical values obtained from following the procedure (*Chem. Lett.* 45, 770–772 (2016)).

Revised Supplementary Table 1:

PC	$E(S_1)$ (eV) ^a	$E(T_1)$ (eV) ^b	τ_{prompt} (ns)	τ_{delayed} (μ s)	$\lambda_{\text{max,abs}}$ (nm)	$\lambda_{\text{max,em}}$ (nm)	$f_{S_0 \rightarrow S_1}$ ^c	Φ_F ^d	Φ_{PF} ^e	Φ_{DF} ^f	Φ_{ISC} ^g	Φ_{RISC} ^h	K_C ⁱ	k_{r,S_1} (10^7 s^{-1})	k_{nr,S_1} (10^7 s^{-1})	k_{ISC} (10^8 s^{-1})	k_{r,T_1} (s^{-1})	k_{nr,T_1} (10^5 s^{-1})	k_{RISC} (10^5 s^{-1})
4DP-IPN	2.55	2.38	2.11	74.67	465	518	0.0750	0.26	0.07 (0.04) ^j	0.19 (0.22) ^j	0.75	1.00	3.71	3.1 ^k	8.9	3.5	-	-	0.53
									(0.04) ^j	(0.22) ^j	0.96	0.89	6.50	1.9 [†]	-	4.6 [†]	-	0.10 [†]	0.77 [†]
4Cz-IPN	2.77	2.68	17.28	4.12	422	524	0.0742	0.64	0.47 (0.24) ^j	0.17 (0.40) ^j	0.26	1.00	1.36	2.7 ^k	1.5	0.15	-	-	3.3
									(0.24) ^j	(0.40) ^j	0.76	0.82	2.67	1.4 [†]	-	0.44 [†]	-	1.2 [†]	5.3 [†]

E_{00} were extracted from the ^aonset of PL and ^bonset of gated PL in ethyl acetate at 65 K, respectively. The onsets were obtained by tangential method, i.e., the intersection of the tangent, set at the high energy slope of the spectrum, with the x-axis. ^cOscillator strengths were obtained by TD-DFT calculation. ^d Φ_F are relatively measured against the coumarin C153.⁹ ^e Φ_{PF} are obtained from the following relationship, $\Phi_{PF} = k_{r,S_1} / (k_{r,S_1} + k_{nr,S_1} + k_{ISC})$. ^f Φ_{DF} are obtained from the following relationship, $\Phi_{DF} = \Phi_F - \Phi_{PF}$. ^g Φ_{ISC} are obtained from the following relationship, $\Phi_{ISC} = k_{ISC} / (k_{r,S_1} + k_{nr,S_1} + k_{ISC})$. ^h Φ_{RISC} are obtained from the following relationship, $\Phi_{RISC} = k_{RISC} / (k_{r,T_1} + k_{nr,T_1} + k_{RISC})$. ⁱ K_C are correlated with the number of TADF cycles determined from the following relationship, $K_C = \Phi_F / \Phi_{PF}$.¹⁰ ^jValues in parentheses of both PCs were determined from the total PLQY (Φ_F) and the proportion of the integrated area of individual components in the TCSPC decay to the total integrated area.^{11,12} ^k k_{r,S_1} was estimated via the Strickler-Berg equation.^{13,14} [†]Photophysical values were obtained following by the procedure^{15,16} under assumption of $k_{nr,S_1} \sim 0$ and $k_{r,T_1} \sim 0$.

In the meantime, since the k_{PET} between PCs and UVAs is estimated by the authors to be around $2.1 \times 10^7 \text{ M}^{-1} \text{ s}^{-1}$, which is around 3 orders higher than either k_{r,T_1} or k_{nr,T_1} . So, in equation 6, calculating T_1 concentration without considering k_{PET} is meaningless to explain why 4DP-IPN is a better PC than 4CZ-IPN (as shown in Fig. R1).

Response: We thank the reviewer for the critical comment. In fact, we used equation 6 solely to obtain photophysical rate constants of PC itself. Then, in the kinetic simulation for the radical generation, we considered all pathways to quench T_1 including electron/energy transfer with co-initiators and UVAs in the mass balance equations (see the revised *Supplementary Table 4*). In response to the reviewer's comment, for the comprehensive understanding of the time evolution of the T_1 concentration, we have revised *Fig. 3b* to include T_1 concentration with consideration of the quenching process with UVAs.

#Revised version of Fig. 3b:

Revised Supplementary Table 4:

Species	Mass balance equations
PC (S ₀)	$\frac{d[S_0]}{dt} = -k_{\text{abs}}(1 - 10^{-\varepsilon \times I \times [S_0]}) + (k_{r,S_1} + k_{nr,S_1})[S_1] + k_{ET,1}[DC][\text{Borate V}] + k_{ET,2}[DA][\text{HNu 254}] + k_{q,1}[T_1][\text{UVA1}] + k_{q,2}[T_1][\text{UVA2}]$
¹ PC* (S ₁)	$\frac{d[S_1]}{dt} = k_{\text{abs}}(1 - 10^{-\varepsilon \times I \times [S_0]}) + k_{\text{RISC}}[T_1] - (k_{\text{ISC}} + k_{nr,S_1} + k_{nr,S_1})[S_1]$
³ PC* (T ₁)	$\frac{d[T_1]}{dt} = k_{\text{ISC}}[S_1] - k_{\text{RISC}}[T_1] - k_{\text{PET},1}[T_1][\text{HNu254}] - k_{\text{PET},2}[T_1][\text{Borate V}] - k_{\text{PET},3}[T_1][\text{DMAEAc}] - k_{q,1}[T_1][\text{UVA1}] - k_{q,2}[T_1][\text{UVA2}]$
² PC ⁺ (DC)	$\frac{d[DC]}{dt} = k_{\text{PET},1}[T_1][\text{HNu254}] - k_{ET,1}[DC][\text{Borate V}]$
² PC ⁻ (DA)	$\frac{d[DA]}{dt} = k_{\text{PET},2}[T_1][\text{Borate V}] - k_{ET,2}[DA][\text{HNu 254}]$
HNu 254	$\frac{d[\text{HNu254}]}{dt} = -k_{\text{PET},1}[T_1][\text{HNu254}] - k_{ET,2}[DA][\text{HNu 254}]$
Phenyl radical (PR)	$\frac{d[\text{PR}]}{dt} = k_{\text{PET},1}[T_1][\text{HNu254}] + k_{ET,2}[DA][\text{HNu 254}]$
Borate V	$\frac{d[\text{Borate V}]}{dt} = -k_{ET,1}[DC][\text{Borate V}] - k_{\text{PET},2}[T_1][\text{Borate V}]$
Boranyl radical (BR)	$\frac{d[\text{BR}]}{dt} = k_{ET,1}[DC][\text{Borate V}] + k_{\text{PET},2}[T_1][\text{Borate V}] - k_{\text{disso.}}[\text{BR}]$
Alkyl radical (AR)	$\frac{d[\text{AR}]}{dt} = k_{\text{disso.}}[\text{BR}]$
DMAEAc	$\frac{d[\text{DMAEAc}]}{dt} = -k_{\text{PET},3}[T_1][\text{DMAEAc}]$
DMAEAc ⁺ (DRC)	$\frac{d[\text{DRC}]}{dt} = k_{\text{PET},3}[T_1][\text{DMAEAc}] - k_{\text{amino.}}[\text{DRC}][\text{DMAEAc}]$
DMAEAc• (DAR)	$\frac{d[\text{DAR}]}{dt} = k_{\text{amino.}}[\text{DRC}][\text{DMAEAc}]$
UVA-1 (UVA1)	$\frac{d[\text{UVA1}]}{dt} = -k_{q,1}[T_1][\text{UVA1}]$
³ UVA-1* (UVA1*)	$\frac{d[\text{UVA1}^*]}{dt} = k_{q,1}[T_1][\text{UVA1}]$
UVA-2 (UVA2)	$\frac{d[\text{UVA2}]}{dt} = -k_{q,2}[T_1][\text{UVA2}]$
³ UVA-2* (UVA2*)	$\frac{d[\text{UVA2}^*]}{dt} = k_{q,2}[T_1][\text{UVA2}]$
Counts of radical generation (RG)	$\frac{d[\text{RG}]}{dt} = k_{\text{PET},1}[T_1][\text{HNu254}] + k_{ET,2}[DA][\text{HNu 254}] + k_{\text{disso.}}[\text{BR}] + k_{\text{amino.}}[\text{DRC}]$

b) I noticed an extra term αI in equation 5 compared to the authors' previous calculations (Adv. Mater., 35 2204776 (2023))? It seems equations 7 and 8 are derived as solutions to equations 5 and 6 without considering the αI term? BTW, what is the A_s in equation 10?

Response: Thanks for the reviewer's careful comments. αI term is related to the extent of photoexcitation to excited states. αI term is not involved to solve ordinary differential equation to obtain the photophysical rate constants. Thus, even some publications of the TADF molecules sometimes omit αI term to describe their differential equations system (e.g., *J. Phys. Chem. A* 125, 8074–8089 (2021), *J. Phys. Chem. C* 122, 29173–29179 (2018)). However, to provide comprehensive understanding of photophysical behaviors, we have revised our descriptions including the photoexcitation term, namely, P_n , fixing some typos like 'As' in equation 10.

#Revised version:

■ Derivation of k_{abs}

Our LED setups are based on 452 nm ($I_0 = 10 \text{ mW cm}^{-2}$) LEDs, therefore with consideration of photonflux ($\text{m}^{-2} \text{ s}^{-1}$), P_n , the rate of S_n generation from S_0 via photoexcitation (i.e., $S_0 \rightarrow S_n$), can be expressed by following equation (3),^{20,21}

$$P_n = \phi \times \frac{A}{V_0 N_A} \times \frac{I_0}{h\nu} \times \epsilon c l \times F, \quad \text{where } F \text{ is photo kinetic factor, } F = \frac{1 - 10^{-\epsilon c l}}{\epsilon c l} \quad (3)$$

where ϕ is quantum yield of the transformation for $S_0 \rightarrow S_n$, A is cross-sectional area (100 cm^2), V_0 is the reaction volume (5 mL), N_A is Avogadro number, h is Planck constant, ν is frequency of the photon, ϵ is the extinction coefficient of PCs in ethyl acetate (e.g., $\epsilon_{452\text{nm}} = 7.8 \times 10^3 \text{ M}^{-1} \text{ cm}^{-1}$ for 4DP-IPN and $\epsilon_{452\text{nm}} = 2.1 \times 10^3 \text{ M}^{-1} \text{ cm}^{-1}$ for 4Cz-IPN), c is the concentration of PC (i.e., $[S_0] = 4.75 \times 10^{-5} \text{ M}$ for the 10 ppm in the synthesis of poly(EHA)) and l is the optical path length ($50 \mu\text{m}$). Because it is tricky to evaluate all the factors affecting photonflux (e.g., refractive index and surface curvature of release film), we excluded them in this kinetic simulation. Furthermore, as the internal conversion (i.e., $S_n \rightarrow S_1$) is highly fast, we would assume the lowest S_1 state are mainly generated. Quantum yield (ϕ) for $S_0 \rightarrow S_1$ is assumed as unity, hence, in accordance with our experimental conditions, equation (3) can be converted to equation (4).

$$P_1 = k_{abs} \times (1 - 10^{-0.005 \text{ cm} \times \epsilon_{452\text{nm}} \times [S_0]}) = 7.6 \times 10^{-4} \text{ M s}^{-1} \times (1 - 10^{-0.005 \text{ cm} \times \epsilon_{452\text{nm}} \times [S_0]}) \quad (4)$$

■ Evaluation of $k_{r,S1}$, $k_{nr,S1}$, k_{ISC} and k_{RISC} .

The general TADF kinetics can be obtained from the differential equations for the singlet and triplet excited state (S_1 and T_1) deactivation.

$$\frac{d[S_1]}{dt} = P_1 + k_{RISC}[T_1] - k_s[S_1] \quad (5)$$

$$\frac{d[T_1]}{dt} = k_{ISC}[S_1] - k_T[T_1] \quad (6)$$

where P_1 is the rate of S_1 generation from S_0 via photoexcitation, $k_s = k_{r,S1} + k_{nr,S1} + k_{ISC}$ and $k_T = k_{r,T1} + k_{nr,T1} + k_{RISC}$; the solutions are described below

$$[S_1] = \frac{[S_1]_{t=0}}{A_1 - A_2} [(k_S - A_2) \exp(-A_1 t) + (A_1 - k_S) \exp(-A_2 t)] \quad (7)$$

$$[T_1] = \frac{[S_1]_{t=0} \cdot k_{ISC}}{A_1 - A_2} [-\exp(-A_1 t) + \exp(-A_2 t)] \quad (8)$$

and the total (i.e. experimentally observed) luminescence time trace is given by²²

$$I(t) = \Phi_F [S_1] + \Phi_{PH} [T_1] \quad (9)$$

$$= \frac{[S_1]_{t=0}}{A_1 - A_2} [(\Phi_F (k_S - A_2) - \Phi_{PH} k_{ISC}) \exp(-A_1 t) + (\Phi_F (A_1 - k_S) + \Phi_{PH} k_{ISC}) \exp(-A_2 t)] \quad (10)$$

The exponents $A_{1,2}$ (which correspond to the reciprocal values of the prompt/delayed PL lifetime constants, i.e., τ_{PF}^{-1} and τ_{DF}^{-1} , respectively) are given by

2. As the authors suggested, the OCA is compatible with the UV curing process, which implies that adding UVAs in color filters would not prevent their UV curing process. However, it would be helpful for readers to better appreciate the rationale behind adding UVAs to the OCA if the authors could discuss the UV-blocking ability of the color filters.

Response: We appreciate the reviewer's comment. To address any potential misunderstanding, we would like to clarify that the conventional production of OCAs using UV-photoinitiators is not compatible in the presence of UVAs due to their high absorbance of UV light. Consequently, we opted for visible-light irradiation, successfully obtaining UV-blocking OCAs. In our revised introduction, we aim to present our argument more clearly and emphasize the potential of UV-blocking color filters.

Previous version:

"This innovative approach necessitates the use of an adhesive material possessing UV-blocking characteristics to replace the polarizer's function of safeguarding the panel from external UV radiation. However, the preparation of UV-blocking adhesive proves to be a complex task due to the UV-blocking agent's high absorbance of UV light, which impedes the effectiveness of the conventional photocuring method that relies on a UV-photoinitiator. Additionally, introducing UV-blocking properties to the color filter might appear viable, but it is impractical for two main reasons: i) these filters are usually produced using a UV curing process,^{9,10} incompatible with the addition of UV absorbers (UVAs), and ii) visible-light curing, while a potential alternative, is ineffective because color filters inherently absorb visible light."

"A simple approach is to utilize visible light instead of UV light for curing, with the aid of a photosensitizer that efficiently absorbs visible light.¹¹⁻²⁶ However, this method presents a challenge as it substantially reduces the film's optical transparency in the visible-light range.²⁷⁻³⁰"

Revised version:

"This innovative approach necessitates the modification of existing layers to possess UV-blocking characteristics to replace the polarizer's function of safeguarding the panel from external UV radiation. To incorporate a UV-blocking ability into the target layer, its fabrication process need to be compatible with UV absorbers (UVAs) considering the potential hindrances of UV-light curing due to their high UV-light absorbance. One strategy would involve imbedding UVAs into the color filter layer. However, since the fabrication of these filters typically relies on UV-light curing,^{9,10} substantial efforts are required to introduce UV-blocking capabilities into the color filters (see below). Optically clear adhesives (OCA) layer could be regarded as an alternative, but the

preparation of UV-blocking adhesive still proves to be a complex task in the presence of UVAs, which impedes the effectiveness of the conventional photocuring method that relies on a UV-photoinitiator.

A simple approach is to utilize visible light instead of UV light for curing, with the aid of a photosensitizer that efficiently absorbs visible light;¹¹⁻³⁰ while visible-light curing can technically be used to add a UV-absorber to the color filter layer, it is likely inefficient and thus impractical, mainly due to the high absorption of visible light by the pigments.^{9,10}

REVIEWERS' COMMENTS

Reviewer #2 (Remarks to the Author):

The authors have provided a detailed explanation of the calculation of the concentrations of [S1] and [T1], which greatly enhances the readers' understanding of their study.

Therefore, I recommend accepting the manuscript.

An optional suggestion is to add a brief discussion of the UV-blocking ability of the color filters regarding my previous comment: "As the authors suggested, the OCA is compatible with the UV curing process, which implies that adding UVAs in color filters would not prevent their UV curing process. However, it would be helpful for readers to better appreciate the rationale behind adding UVAs to the OCA if the authors could discuss the UV-blocking ability of the color filters." I understand that the conventional production of OCAs using UV photoinitiators is not compatible with the presence of UVAs due to their high absorbance of UV light. However, is it possible that the color filters have a good UV absorbance or blocking ability?

Reviewer #2 (Remarks to the Author):

The authors have provided a detailed explanation of the calculation of the concentrations of [S1] and [T1], which greatly enhances the readers' understanding of their study. Therefore, I recommend accepting the manuscript.

Response: We sincerely appreciate the very positive response from the reviewers. In this revised version, we have addressed reviewer's remaining comments, aiming to provide readers with a more comprehensive understanding of our work. We highlighted our revision in main manuscript as blue color. We are grateful for the continued efforts that the reviewers have dedicated to our work to significantly improve its quality.

An optional suggestion is to add a brief discussion of the UV-blocking ability of the color filters regarding my previous comment: "As the authors suggested, the OCA is compatible with the UV curing process, which implies that adding UVAs in color filters would not prevent their UV curing process. However, it would be helpful for readers to better appreciate the rationale behind adding UVAs to the OCA if the authors could discuss the UV-blocking ability of the color filters." I understand that the conventional production of OCAs using UV photoinitiators is not compatible with the presence of UVAs due to their high absorbance of UV light. However, is it possible that the color filters have a good UV absorbance or blocking ability?

Response: Thanks for the reviewer's comments. Color filter contains pigments or dyes of which optical properties can be tuned by change of substituents enabling light absorption at specific wavelengths (*Dyes Pigm.* **88**, 166–173 (2011), *New J Chem.* **36**, 812–818 (2012), and *Dyes Pigm.* **154**, 128–136 (2018)). As the reviewer suggested, to elucidate our novelty and avoid the reader's misunderstanding, we revised the introduction with sentences on the feasibility in the UV-blocking properties of the color filter layer.

Previous version:

"To incorporate a UV-blocking ability into the target layer, its fabrication process need to be compatible with UV absorbers (UVAs) considering the potential hindrances of UV-light curing due to their high UV-light absorbance. One strategy would involve imbedding UVAs into the color filter layer. However, since the fabrication of these filters typically relies on UV-light curing,^{9,10} substantial efforts are required to introduce UV-blocking capabilities into the color filters (see below)."

Revised version:

"To incorporate a UV-blocking ability into the target layer, its fabrication process need to be compatible with additives which effectively absorb the UV radiation such as pigments or UV absorbers (UVAs). One approach to introduce these capabilities would involve fine-tuning the pigments in the CF layer, but it would require endeavors to ensure they absorb simultaneously UV light and visible light at desired wavelengths.⁹⁻¹¹ An alternative strategy would be imbedding UVAs into the CF layer. However, since the fabrication of these filters typically relies on UV-light curing,^{12,13} substantial efforts are needed to overcome potential hindrances of UV-light curing associated with their high UV-light absorbance."